# Strengthening gradients in the tropical west Pacific connect to European summer temperatures on sub-seasonal timescales

Chiem van Straaten[1,2], Dim Coumou[1,2,3], Kirien Whan[1], Bart van den Hurk[2,4], and Maurice Schmeits[1,2]

[1]Royal Netherlands Meteorological Institute (KNMI), De Bilt, the Netherlands
[2]Institute for Environmental Studies (IVM), Vrije Universiteit Amsterdam, Amsterdam, the Netherlands
[3]Institut Pierre-Simon Laplace, CNRS, Université Paris-Saclay, Sorbonne Université, France
[4]Deltares, Delft, the Netherlands

**Correspondence:** Chiem van Straaten (j.w.vanstraaten@vu.nl)

**Abstract.** Recent work has shown that (sub-)seasonal variability in tropical Pacific convection, closely linked to ENSO, relates to summertime circulation over the Euro-Atlantic. The teleconnection is non-stationary, probably due to long-term changes in both the tropical Pacific and extra-tropical Atlantic. It also appears imperfectly captured by numerical models. A dipole in west Pacific sea surface temperatures (SSTs) was found to be the best predictor of errors in numerical sub-seasonal forecasts of European temperature. In this diagnostic study we use reanalysis data to further investigate the teleconnection pathway and the processes behind its non-stationarity. We show that SST gradients associated with the dipole represent a combination of ENSO variability and west Pacific warming, and have become stronger since 1980. Associated patterns of suppressed and enhanced tropical heating are followed by quasi-stationary waves that linger for multiple weeks. Situations with La Niña-like gradients are followed by high pressure centers over eastern Europe and Russia, three to six weeks later. Inverted situations are followed by high pressure over western Europe, three to six weeks later. The latter situation is conditional on a strong meridional tripole in north Atlantic SST and a co-located jet stream. Overall, the sub-seasonal pathway diagnosed in this study connects to patterns detected at seasonal scales, and confirms earlier findings that the summertime connectivity between the Pacific and Europe has shifted in recent decades.

## 1 Introduction

Sub-seasonal forecasts are made with a lead time of 2-6 weeks. For weather at any mid-latitude location, part of the predictability at that lead time originates in the tropics (Vitart and Robertson, 2018). Tropical deep convection and associated diabatic heating generate upper level divergence and vorticity anomalies that force Rossby Waves (Hoskins and Karoly, 1981; Sardeshmukh and Hoskins, 1988; Trenberth et al., 1998). These waves can propagate into the westerly mid-latitude flow and steer associated weather patterns, commonly affecting the mid-latitudes beyond two weeks into the future (Liu and Alexander, 2007; Branstator, 2014; Stan et al., 2017). Impacts on mid-latitude surface weather are especially pronounced when waves are quasi-stationary (Schubert et al., 2011; Wolf et al., 2018; Röthlisberger et al., 2019). An example of a teleconnection from tropical heating is the Madden Julian Oscillation, which in the Euro-Atlantic sector, has a phase-dependent influence on the North Atlantic Oscillation (Cassou, 2008; Henderson et al., 2017; Vitart, 2017).

Since its origins, teleconnection research has mostly focused on winter, as this is a season with higher baroclinicity and more potent Rossby wave propagation (Bjerknes, 1969; Trenberth et al., 1998; Branstator and Teng, 2017). Teleconnections are also thought to have an influence in summer (Cassou et al., 2005). Heating patterns over the western tropical Pacific are of primary importance for summertime quasi-stationary Rossby waves (QSRWs) (Ting, 1994; Behera et al., 2013; Ma and Franzke, 2021). The dipole of enhanced convective activity over the Maritime continent, in conjunction with reduced activity over the west and central Pacific is related to circulation over the Euro-Atlantic sector (O'Reilly et al., 2018; Fuentes-Franco and Koenigk, 2020). One feature known for producing such zonal contrasts in sea surface temperature (SST) and atmospheric heating, is the El Niño Southern Oscillation (ENSO). When ENSO's atmospheric component (the Walker circulation) strengthens, convection over the Maritime continent increases and that over the tropical central Pacific decreases (Bjerknes, 1969). It is for instance thought that a (developing) La Niña episode supported the prominent blocking that was part of the Russian heatwave of 2010 (Schneidereit et al., 2012). On the other hand, it is also thought that SSTs had little influence on the Russian heatwave (Dole et al., 2011; Hauser et al., 2016; Wehrli et al., 2019).

Besides the zonally oriented dipole implicated in ENSO, namely between the tropical central Pacific and Maritime continent, meridional orientation is important as well. Diagnosis by Ding et al. (2011) of heating patterns that steer mid-latitude flow reveals that anomalous subtropical heating can relate to QSRWs. Such meridional features reflect activity of the Indian Summer Monsoon and the western north Pacific monsoon, as both consist of anomalous convection extending northward, into the Indian subcontinent and the western north Pacific (WNP) respectively. SST contrasts in both the zonal and meridional direction have increased rapidly since 1990 by what is called the 'west Pacific warming mode' (Funk and Hoell, 2015). This long term change is diagnosed in SSTs from which the first order effects of ENSO have been removed and can thus be viewed as change in the 'background state' of the Pacific. It consists of concentrated warming over the Maritime continent and the WNP region. This is found to be a response to anthropogenic emissions (Funk and Hoell, 2015) and has strengthened the Walker circulation (Funk et al., 2018; Lee et al., 2022).

Coinciding with west Pacific warming, observed connectivity between the Pacific and Euro-Atlantic circulation appears to have strengthened (O'Reilly et al., 2019; Sun et al., 2022). This strengthening is not a consequence of internal amospheric variability but is a response to the SST trends (O'Reilly et al., 2019), potentially influencing current and future Euro-Atlantic circulation. Unfortunately, numerical climate models seem unable to reproduce the observed changes in Euro-Atlantic circulation (Boe et al., 2020; Vautard et al., 2023), meaning they underestimate the rapid increase of European summer temperature extremes (van Oldenborgh et al., 2022). Also numerical weather prediction (NWP) models have shortcomings in simulating Rossby Wave teleconnections (O'Reilly et al., 2018; Quinting and Vitart, 2019). We previously found that west Pacific SSTs could relate to QSRWs that conditionally led to monthly European summer temperature anomalies. This teleconnection appeared imperfectly represented in the numerical model of the European Center for Medium-range Weather Forecasts (ECMWF) (van Straaten et al., 2023).

The shortcomings of weather- and climate-models can reside in many processes associated with Rossby Wave teleconnections. Whether QSRWs influence a remote location can be modulated by processes related to the forcing of waves, and processes related to wave propagation and amplification (White et al., 2022). In the Pacific region, or further along the propa-

gation trajectory, other sources of heating and vorticity can strengthen the QSRW when they are in-phase, and negate it when they are out-of-phase, dependent on the optimal forcing pattern of the QSRW (Schubert et al., 2011; Kim and Lee, 2022). Extra-tropical Pacific SST anomalies provide such feedback when preceding dynamics have left in place an anomaly pattern that is in-phase with a summertime QSRW (Vijverberg and Coumou, 2022). In the same way the state of the north Atlantic can permit or hinder QSRW propagation towards Europe (Fuentes-Franco et al., 2022). Also soil moisture depletion, for example over the US, can amplify wave patterns and make them circumglobal (Teng and Branstator, 2019).

A second potential modulator is the way atmospheric jets function as waveguides (Hoskins and Ambrizzi, 1993; White et al., 2022). Waveguides are sharp gradients in the background flow, along which Rossby Waves propagate (Wirth et al., 2018; Manola et al., 2013). The role of jets is noticeable as strong waves often emanate at their exit regions, of which the Euro-Atlantic sector is one (Stan et al., 2017). The characteristics of the background flow, in the form of jet position, width and strength can determine whether a wave response will be of limited longitudinal extent, or circumglobal (Branstator and Teng, 2017).

In this study we characterize the west Pacific teleconnection to European summer temperature variability and evaluate whether the teleconnection relates long term changes in the Pacific to Euro-Atlantic circulation. To that end we build an index for an SST dipole over the west Pacific, and an index for European surface temperature more than two weeks later. Such a lagged, sub-seasonal timeframe is different from the concurrent, seasonal diagnostics used in earlier studies (Ding et al., 2011; Behera et al., 2013; O'Reilly et al., 2018). We assess the apparent strengthening of teleconnection and investigate whether it is modulated by (a combination of) the processes described above. Overall, we hope that our diagnostic framework can aid the inspection of this teleconnection in weather- and climate-models, such that long term projections and sub-seasonal forecasts of European summer extremes can ultimately be improved.

## 2 Data

In this study we use the ERA5 reanalysis (Hersbach et al., 2020). We extract daily values of sea surface temperature (SST) and two-meter temperature (t2m), as they will form the respective start- and end-point of the teleconnection. Top-of the atmosphere outgoing longwave radiation (OLR) was extracted as indicator of tropical deep convection, zonal wind at 300 hPa (u300) as an indicator of jet stream strength and position, and geopotential height at 300 hPa (z300) as an indicator of the QSRW itself. Values were extracted from 1950 until 2021, in a domain that spans from 20$^{\circ}$S to 90$^{\circ}$N, and 180$^{\circ}$W to 180$^{\circ}$E, at a spatial resolution of 0.25x0.25$^{\circ}$.

Daily values were de-seasonalized by approximating the seasonal cycle with a polynomial that is a function of the day-in-the year (as in Mayer and Barnes, 2021, 2022). We found that a 7-degree polynomial was most suited to accomodate the changing shape of seasonal cycles with latitude. At polynomials below 5 degrees the residuals remained visibly dependent on the season. For each grid cell the polynomial was fitted to the complete set of years from 1950 till 2021. Daily anomalies of all variables were then averaged to four-week (28-day) values. In previous studies we namely found that the four-week or 'monthly' average t2m is predictable with sub-seasonal lead times (van Straaten et al., 2022, 2023). The aggregation is executed as a rolling-window averaging, such that one value was recorded each day.

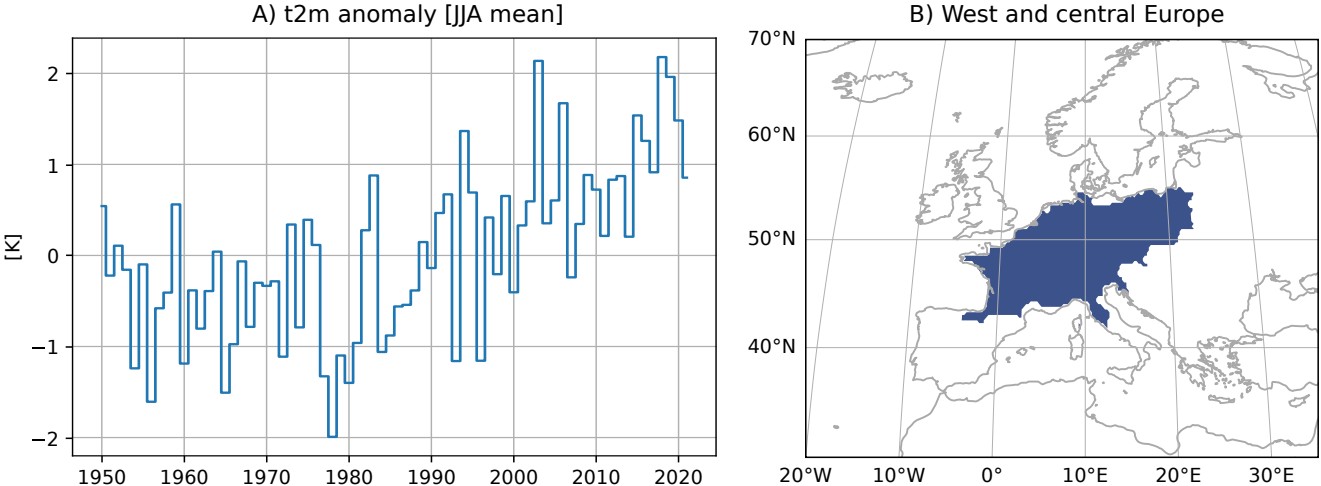

**Figure 1.** Two-meter temperature (t2m) anomaly in west and central Europe. A) Seasonal mean (JJA) ERA5 t2m anomaly in the region, from 1950 to 2021. B) west and central European region.

The western north Pacific (WNP) region, whose warming as part of the west Pacific warming mode was found to have strong influence on the Walker Circulation, is defined from 10ºN–30ºN 130ºE–170ºW (Funk et al., 2018). In this region we record each day the spatial mean, four-week average SST anomaly.

To investigate the role of ENSO we use pre-computed monthly relative ENSO indices (van Oldenborgh et al., 2021), based on the ersstv5 dataset of ocean observations (Huang et al., 2017), in regions Niño 3 (5ºS–5ºN, 90-150ºW) and Niño 4 (5ºS–5ºN, 160ºE–150ºW) (Trenberth and Stepaniak, 2001). Relative ENSO indices are computed like regular ENSO indices as the average SST anomaly within a Niño region (climate normals defined from 1981-2010), but from which the average SST anomaly between 20ºN and 20ºS is subtracted. This makes the indices less distorted by global warming and more useful for 100 describing teleconnections (van Oldenborgh et al., 2021). We interpolate them linearly to obtain a monthly value each day.

For the Pacific Decadal Oscillation (PDO) we use standardized values of the first principle component of monthly Pacific SST anomalies north of 20ºN (Mantua et al., 1997). The pre-computed index is provided by NOAA and is also based on ersstv5. We interpolate the PDO index linearly to obtain a monthly value each day.

As mentioned, we previously found that the average t2m anomaly is predictable when aggregated to the four-week or 105 'monthly' time scale (van Straaten et al., 2022). Crucially, this predictand involved spatial averaging as well. Sub-seasonal predictions for individual grid cells are usually less successful, because only the overarching variability that affects daily anomalies at multiple moments and locations can be predictable (Buizza and Leutbecher, 2015; Wheeler et al., 2017; van Straaten et al., 2020). In the mentioned previous study we used hierarchical clustering to find a west and central European region (Fig. 1B), in which the average t2m anomaly is predictable (van Straaten et al., 2022). For the remainder of this study 110 we refer to this predictand as 'Western European t2m in week 3,4,5 and 6'. A diagnostic plot of June-July-August (JJA) averages of this variable shows that summer temperatures have been warming (Fig. 1A). In fact, western European summer

temperatures have been warming faster than the global average, especially since the 1990's, at a rate of 0.4 to 0.8 °C/decade (Christidis et al., 2015; Dong et al., 2017; Gutiérrez et al., 2021).

## 3   West Pacific dipole index

Figure 2 displays Pacific SST grid-cells whose variability precedes European t2m by more than two weeks. We correlate four-week-averaged SST anomalies ('SST in week -3 to 0') to the lagged European four-week-averaged response ('t2m in week 3 to 6'). Our use of rolling averages leads to 92 samples per JJA season. The gap of two weeks between the two periods corresponds to the time window over which tropical Rossby Waves still affect the mid-latitudes (Branstator, 2014) (alternative lags are discussed in Appendix A). The presented correlation is corrected for inflating factors like global warming and auto-

correlation (Fig. 2A). Specifically, we compute the partial correlation between residual SST and residual t2m. First we let a linear regression predict observed SST and t2m anomalies using time and the value of the previous time step (details can be found in van Straaten et al., 2022). These predictions are then subtracted from the observed anomalies, resulting in residual SST and t2m. With the confounding effects of global warming and auto-correlation removed, any correlation that remains significant is more likely to represent a sub-seasonal relation. Significance of the partial correlation is determined per grid cell

by a two-sided test with nominal level $\alpha = 5 \cdot 10^{-12}$ (this small $\alpha$ was a pragamatic way of accounting for the dependence introduced by rolling window averaging). To mitigate the accumulation of chance-based discoveries when performing multiple significance tests, we applied a false discovery rate correction (Benjamini and Hochberg, 1995) (details can be found in van Straaten et al., 2022). Within the full dataset of 1950 to 2021, we further test robustness of the partial correlation with a five-fold crossvalidation, leaving out consecutive blocks of 14 years. Grid cells with correlations significantly different from zero in five

out of five subsets are highlighted in yellow (Fig. 2B).

The highlighted SST regions relate to European t2m, and do not correspond perfectly to patterns that explain the highest amount of Pacific variability, like PDO and ENSO. Aspects are however captured. A large cluster of significant cells resides in the region used to define the PDO (Mantua et al., 1997) (Fig. 2B). Also at the western edge of the Niño 4 area we see a cluster of cells (called 'component 1' (1.5ºS–5.5ºN, 162-169ºE)). This cluster is flanked by 'component 2' (10–14ºN,140-150ºE)

which lies within the WNP area (Fig. 2). These regions were the focus of the study by van Straaten et al. (2023), showing that the opposed sign of anomalies in these two regions, when captured with a spatial co-variance predictor, were important and related to errors in the ECMWF model. The respective positive and negative correlations of component 1 and 2 (Fig. 2A), hint at situations with anomalously warm SSTs in component 1 and anomalously cold SSTs in component 2, and in which week 3 to 6 European t2m would later be above normal (and vice-versa). This emphasizes the importance of the western Pacific

in forcing a teleconnection towards Europe (as seen in Ding et al., 2011; Behera et al., 2013; O'Reilly et al., 2018; Ma and Franzke, 2021). Furthermore, with component 2 located in the WNP and component 1 at the eastern edge of the central Pacific, their combination appears to capture a heating contrast with both meridional and equatorial orientation (i.e. the emphasized signal consists of more than just zonally opposing anomalies along the equatorial axis).

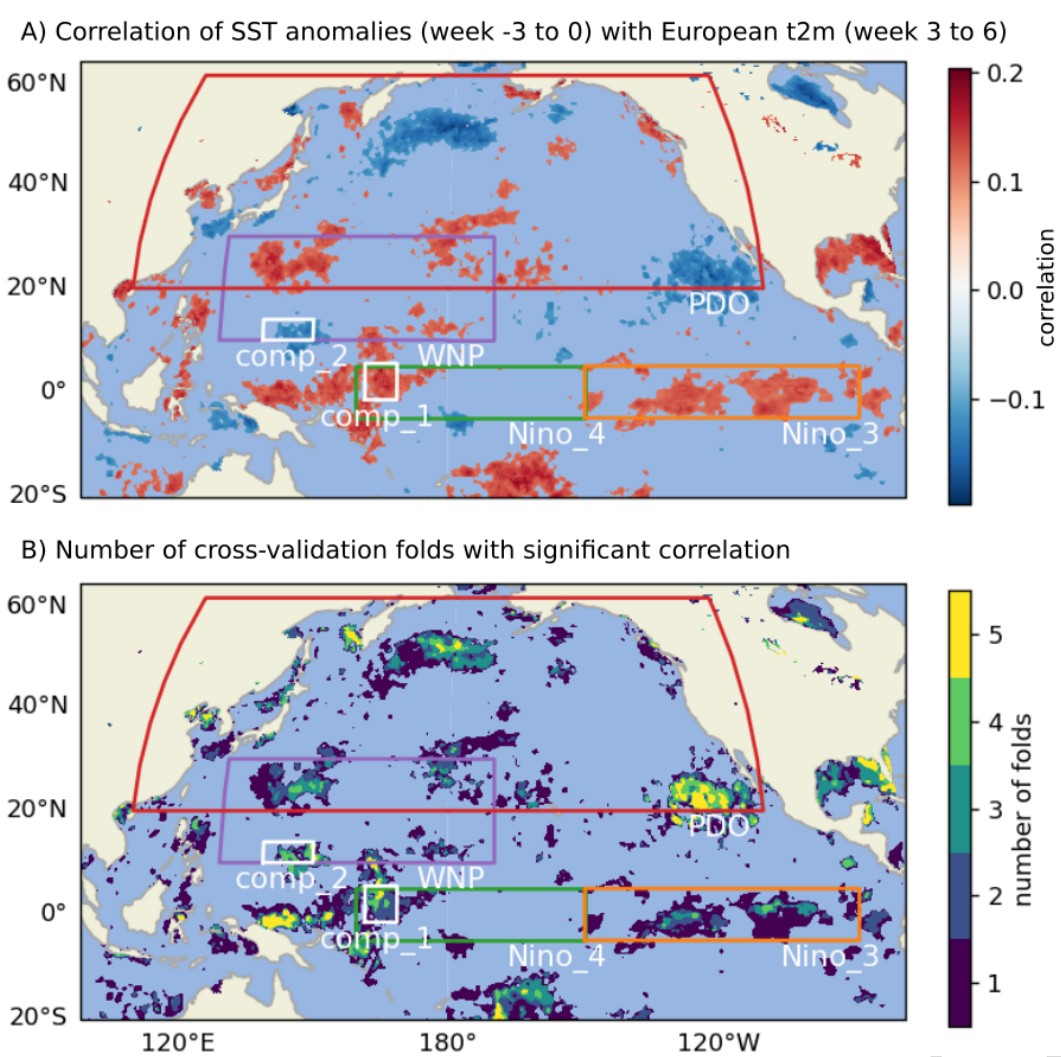

**Figure 2.** Connection between ERA5 Pacific SST anomalies and European t2m anomalies at sub-seasonal timescales in summer (1950-2021). A) Spearman rank correlation between week -3 to 0 SST anomalies and week 3 to 6 western European t2m, corrected for inflation by linear trends, seasonality and auto-correlation. Reported is the mean correlation over 5 crossvalidation folds, constructed by leaving out consecutive blocks of 14 years. B) Robustness of the correlation as measured by the number of folds with significant correlation. Annotated are regions commonly used to capture Pacific variability. The two components of the west Pacific Dipole (WPD) index are highlighted by white squares.

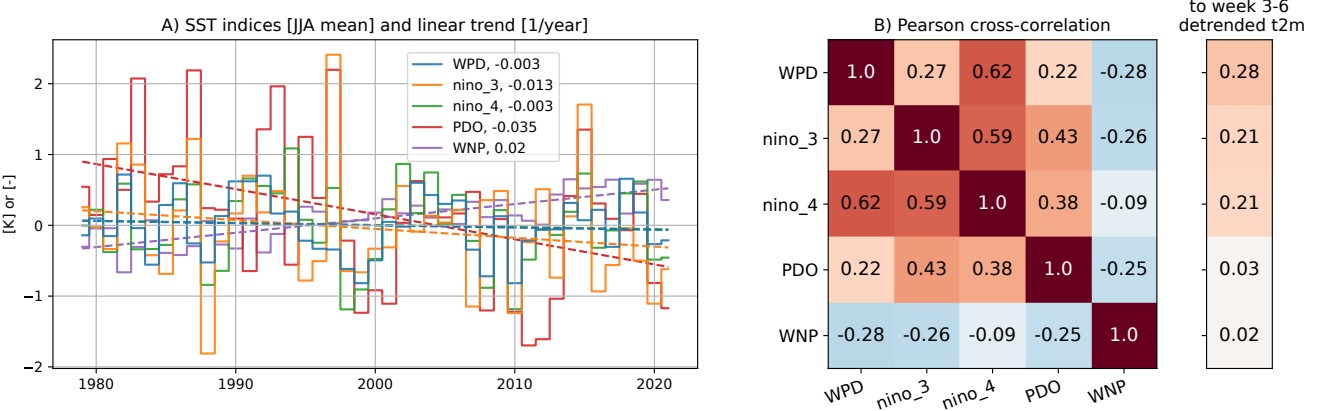

**Figure 3.** Correspondence between the west Pacific Dipole index (WPD: SST in 'component_1' minus SST in 'component_2', Fig. 2), and other Pacific indices. A) Seasonal mean timeseries and linear trend from 1979 till 2021. B) Cross-correlation matrix between the SST indices (JJA only). C) Lagged correlation between SST indices and European t2m in week 3 to 6, corrected for a linear trend in t2m. The SST indices capture SSTs from week -3 to 0 (see also section 2). Shading in panels B and C illustrates the sign of the reported correlation values (red: positive, blue: negative) and their magnitude (dark: strong, light: weak).

Here we simply capture this contrast by defining a west Pacific Dipole (WPD) index, namely the four-week anomaly in
component 1 minus the four-week anomaly in component 2. As SSTs in component 2, which is further west in the Pacific, are, in absolute terms, generally warmer than those in component 1, this definition dictates that: positive WPD values represent a weaking of the climatological SST gradient, and negative WPD values represent a strengthening of the climatological gradient. The reason that positive WPD is defined as 'positive', is its association to above-normal western European t2m (Fig. 3C).

To test the sensitivity of choices regarding the location of the two boxes, we present additional results in Appendix A. These
are additional crossvalidation maps, similar to Fig. 2B, and show that the exact extent and location of the robustly correlated pattern can shift when different combinations of timescales and lags are chosen. The current boxes are positioned such that only the features shared among multiple combinations are captured.

We expect the WPD index to relate to other Pacific modes. PDO for instance, comprises a combination of remote ENSO-induced variability in combination with local atmosphere-ocean interactions (Newman et al., 2016). Especially due to spatial
overlap in regions, we see that WPD relates to Niño 4 and WNP, which is illustrated by cross-correlations of 0.61 and -0.29 respectively, in the period from 1979 until 2021 (Fig. 3B). (We do not include the pre-1979 period because it lacks the west Pacific warming mode and the stronger Pacific-Atlantic connections (Funk and Hoell, 2015; O'Reilly et al., 2019), which would muddy the illustration of current inter-relations.) The real difference with for instance Niño 4, is that the WPD-index is designed to specifically target the teleconnection towards Europe, which is confirmed by the highest correlation of all indices
to week 3-6 t2m (Fig. 3C).

Correlation between Pacific modes is further illustrated by a year-to-year representation of the time series (Fig. 3A). Over the 1979-2021 period the standardized PDO is moving towards a negative phase (-0.031 std/yr). Also Niño 3 displays a slight negative trend (-0.013 K/yr), which is different from Niño 4 and WPD, and opposed to the positive trend in WNP (0.02 K/yr). These opposing trends reflect documented warming in the western Pacific, while the central to eastern tropical Pacific has not warmed (Wills et al., 2022; Seager et al., 2022; Sun et al., 2022; Lee et al., 2022). The result is that the zonal SST gradient over the tropical Pacific has increased, which is generally referred to as a 'La-Niña-like' change in the Pacific 'background state', and has been linked to a stronger Walker circulation (Lee et al., 2022; Seager et al., 2022). Also the west Pacific warming mode, visible as the warming in the WNP region, enhances the climatological background-gradient over the Pacific, and is linked to a strengthening of the Walker circulation (Funk and Hoell, 2015; Funk et al., 2018), with potentially important influences on tropical-extratropical teleconnections (Schubert et al., 2014; O'Reilly et al., 2019; Sun et al., 2022).

## 4 Emergence of a teleconnection

We classify the WPD index into three tercile-based categories. The positive phase is when SSTs during week -3 to 0 are anomalously warm in component 1 and anomalously cold in component 2. In the negative phase this is inverted, and a neutral phase is in-between. Because the WPD index does not have a significant trend (Fig. 3A, -0.003 K/yr), we determine tercile thresholds over the entire 1950-2021 dataset. Week 3-to-6 t2m on the other hand, shows a clear trend (Fig. 1A). This trend can be due to a combination of thermodynamic and dynamic processes that can be both of forced and/or unforced origin (see e.g. Deser et al., 2016; Faranda et al., 2023). Since this study focuses on the sub-seasonal connection between WPD and Euro-Atlantic circulation we remove the long term warming trend by computing t2m tercile thresholds per rolling time window of 21 summers. If not, the upper tercile 'positive' t2m class would mostly consist of samples from recent years. With the distortion removed we can define our measure of teleconnection strength or 'sub-seasonal connectivity' as: the number of occasions with (i) *positive WPD* AND *positive t2m response*, plus the number of occasions with (ii) *negative WPD* AND *negative t2m response*. A window length of 21 summers was deemed sufficiently short to adapt to non-stationarity in the roughly 72-year dataset, but sufficiently long not to be affected by inter-annual variability.

Temporal evolution of the WPD classes reveals that since 1950 (the beginning of the dataset) the negative WPD phase has strongly increased in frequency (blue line in Fig. 4A). Until 2000 this happens mostly at the expense of the neutral phase, and after that also at the expense of the positive phase. The dwindling occurrence of the neutral phase can explain why the tercile distribution shows large changes (Fig. 4A), whereas the seasonal mean of WPD had no signficant trend (Fig. 3A). The increased occurrence of negative WPD phases does agree with WNP warming and the strengthening of climatological gradients over the Pacific (Section 3, Fig. 3A).

Concurrent with this development we see that lagged t2m phases in week 3 to 6 increasingly follow the WPD phases in week -3 to 0. The sum of positive and negative responses to respectively, positive and negative WPD, rises beyond values found by chance (grey area, Fig. 4B). This roughly occurs for 21-year windows centered in 1990 and beyond, meaning it starts in 1980. The emergence of a significant teleconnection between the Pacific and Euro-Atlantic circulation is reported by other studies,

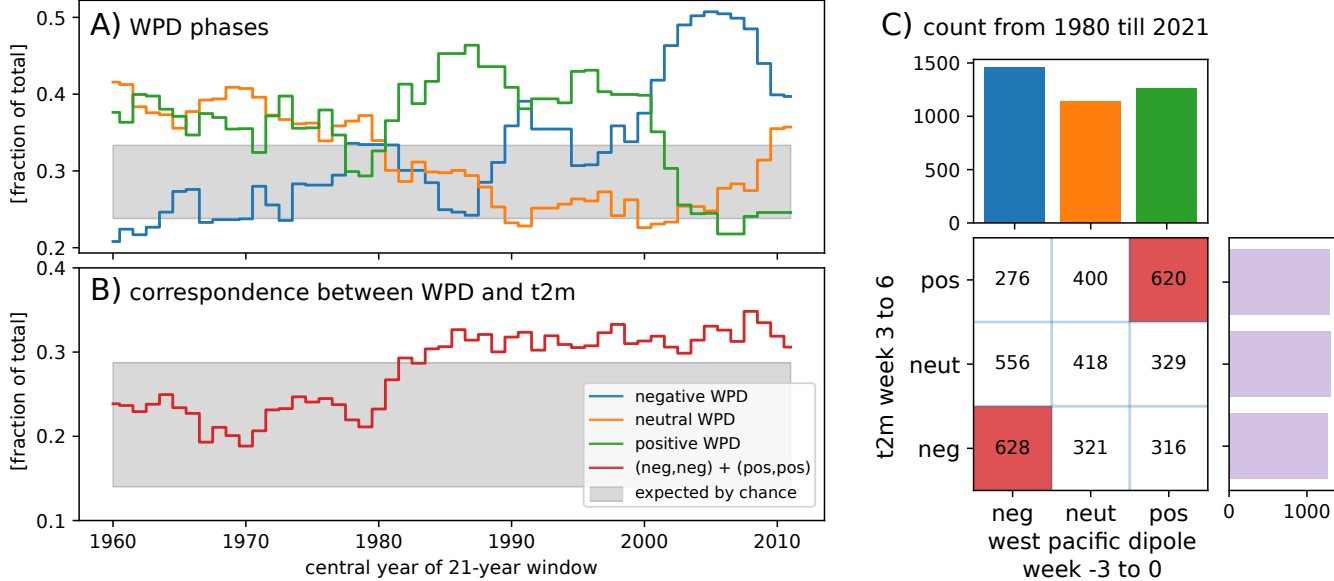

**Figure 4.** Teleconnection between week -3 to 0 west Pacific Dipole index and week 3-6 European t2m, as measured by correspondence in tercile classes (negative, neutral, positive). A) Prevalence of the three WPD classes in a moving window of 21 summer seasons (blue: negative WPD; orange: neutral WPD; green: positive WPD), with tercile thresholds determined over the entire dataset, from 1950-2021. Presented as fraction of the total number of samples within each 21-year rolling window. B) Summed occurrence of the negative and positive phase of the teleconnection, i.e. a negative WPD preceding a negative t2m response plus a positive WPD preceding a positive t2m response, as counted for each 21-season window, whereby the t2m tercile thresholds are re-computed in each window. Grey areas in (A) and (B) denote the 0.025 and 0.975 quantiles of the uncertainty distribution when counting is performed on 21 summer seasons that are randomly sampled from the 1950-2021 dataset (500 repeats, with replacement). C) Distribution of tercile classes for WPD (x-axis) and European t2m (y-axis) as counted for each 21-season window from 1980 till 2021 (central years 1990-2011) when the teleconnection (both in negative and positive phase) was present in a statistically significant way (as shown in panel B).

both for the past decades (Wu and Lin, 2012; Lim et al., 2019; O'Reilly et al., 2019; Sun et al., 2022), and for the near-future
(Mayer and Barnes, 2022).

In the remainder of this paper we focus on the period 1980-2021 in which a significant teleconnection has emerged. The joint distribution of WPD and t2m phases is given in Figure 4C. Presented are the number of samples in each category (because of rolling averaging one sample per summer day). Note that the marginal distribution of t2m is uniform, because tercile thresholds are re-estimated in each rolling 21-year window, and that the marginal of WPD is not uniform and shows the relative dominance
of negative WPD phases in this period.

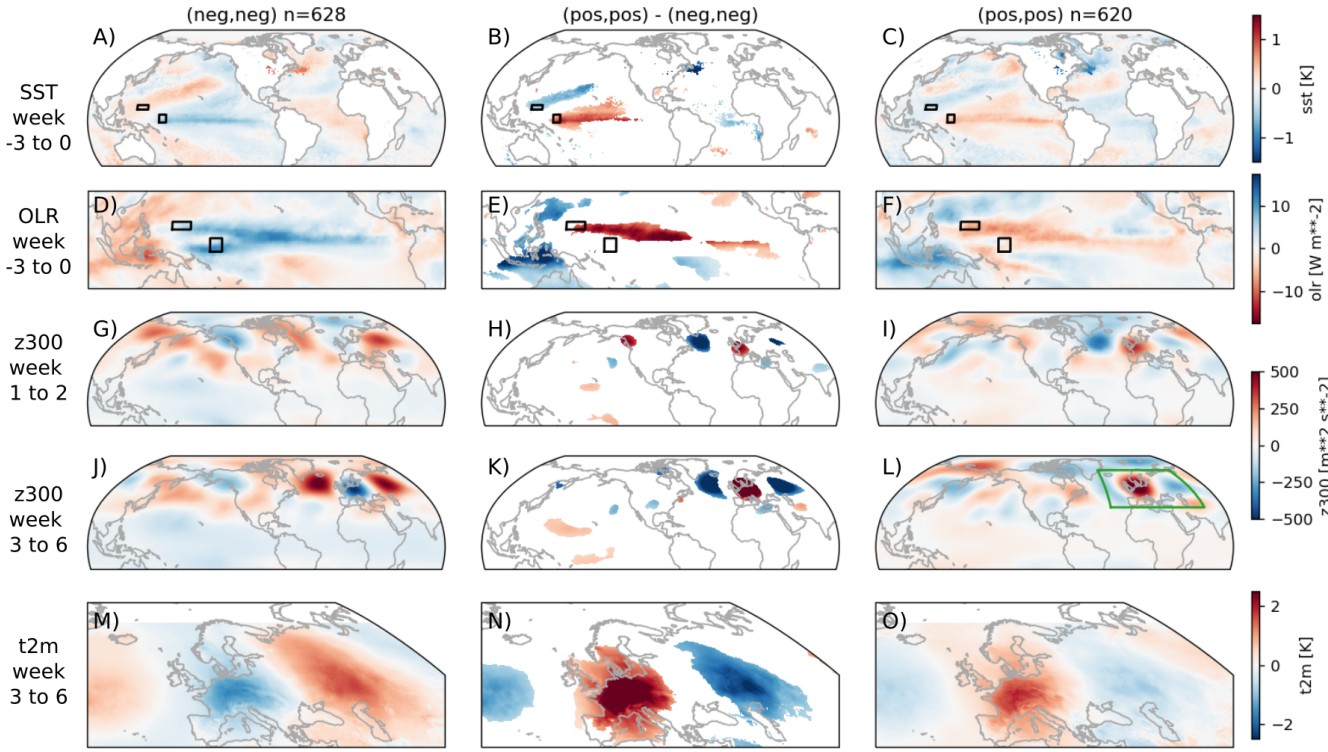

**Figure 5.** Composite plots illustrating the teleconnection between week -3 to 0 west Pacific SST and week 3 to 6 west-central European t2m, based on composite anomalies from the period 1980-2021 (climate normal also estimated from 1980-2021). Left column: Samples in which negative WPD phases precede the negative t2m class (n=628). Right column: Samples in which positive WPD phases precede the positive t2m class (n=620). Middle column: Difference between panels in the right and left column. Values found to not exceed the difference as expected by random chance are masked ($\alpha = 0.05$, more detail in Section 5). From top to bottom: SST pattern and OLR in week -3 to 0, the subsequent z300 response in week 1 and 2, and the eventual impact on z300 and surface temperature in week 3 to 6. Black insets in panels A-F show the two components of the West Pacific Dipole index. Green inset in panel L serves as reference for the spatial extent of surface t2m in the bottom row.

## 5 Forcing, quasi-stationary wave, and surface imprint

We now investigate the spatial and temporal patterns that occur during the period that the teleconnection is significant. Different components of the pathway (i.e. SST, OLR, z300, t2m) are plotted as rows in Figure 5. The negative phase, positive phase and their difference are plotted in the left, right and middle column, respectively. Unmasked grid cells in the middle column display differences that were found to be significantly different from differences that can arise by chance. To conduct this two-sided test we resampled entire summer seasons from the 1980-2021 data, with replacement (i.e. a block-bootstrap), until groups with sizes as in Fig. 4C were obtained. We then computed the difference between their composite means and repeated the procedure 500 times to obtain a sampling distribution per grid cell. Subsequently we tested whether the actual difference fell outside the distributions' empirical quantiles (q0.025 and 0.975 for $\alpha = 0.05$, and q0.05 and 0.95 for $\alpha = 0.1$).

It is clear that the WPD index captures a large geographical pattern with pronounced SST and OLR anomalies accross the Pacific ocean, despite being defined by small boxes (Fig. 5A-F). Visually, the SST states in week -3 to 0 represent a combination of three patterns: (i) ENSO, which is visible in the La-Niña-like equatorial contrast between anomalously cool SSTs in Niño 4 and anomalously hot SSTs around the Maritime continent in Fig. 5A, and in the El-Niño-like contrasts in Fig. 5C, (ii) the west Pacific warming mode, which Funk and Hoell (2015) describe as a tilted 'V' that connects the above-normal SSTs around the Maritime continent to the extra-tropics, in north-eastward and south-eastward direction (the north-eastward extension from component 2 is significantly present in Fig. 5B), and (iii) a PDO-like pattern consisting of the same north-east extension of warm SST anomalies, surrounded by cold anomalies to its south, west and north, which is a Pacific configuration known to provide summertime predictability for the eastern US (our Fig. 5A and Fig. 1F of Vijverberg and Coumou (2022)). Generally, patterns of SST anomalies captured by WPD resemble the results of the mentioned studies only partially. For instance, the 'cold surrounding', which is part of the PDO-like pattern, does not show up as statistically significant in this study. Such differences are understandable because the WPD index is designed for direct sub-seasonal association to summertime European t2m, which, only to a lesser degree, is found in the other modes (Fig 3C). The important WPD features, emphasized by significance testing, are the tropical central-to-west Pacific SST anomalies, and the ones in the WNP region, extending north-eastward from component 2 (Fig. 5B).

In OLR we see that the WPD phases correspond to inverted patterns of anomalous deep convection, which need not be aligned with the dipole in anomalous SST and the WPD boxes themselves (Fig. 5D-F). Like SST however, the heating patterns during weeks -3 to 0 consist of an opposition with both equatorial and meridional orientation. Focusing for instance on positive WPD phases, the heating over the Maritime Continent and in the Western North Pacific monsoon region (north-east of the Philippines) is reduced, and the heating over the central Pacific is enhanced (Fig. 5E-F). This opposition matches the first mode of summertime tropical precipitation variability that was found earlier to be important for steering mid-latitude circulation (Ding et al., 2011). The main difference with the analysis of Ding et al. (2011) is that their analysis diagnosed a concurrent seasonal link, whereas our results suggest that it consists of an underlying, lagged and sub-seasonal relation (see also Appendix A).

The forcing in weeks -3 to 0 translates into an initial atmospheric response in weeks 1-2 (Fig. 5G-I). Centres of action (locations with the largest z300 anomalies) are found from the north Pacific to Eurasia, showing the considerable longitudinal extent of the QSRW (Fig. 5H). Characterisic and significant centres are the high pressure anomalies in north-west US and the low pressure anomalies situated south of Greenland and west of the British Isles (Fig. 5I), and which are inverted in the negative phase (Fig. 5G). These centres are part of summertime patterns diagnosed both with a focus on the US heatwaves (Vijverberg and Coumou, 2022) and with a focus on the Euro-Atlantic circulation (Wulff et al., 2017; O'Reilly et al., 2018). As the latter studies used variance decomposition methods (in contrast to our WPD- and t2m-based compositing), the found centres appear to be robust across methods. An interesting difference is that the tropical forcing of the south of Greenland cyclonic anomaly was thought to come primarily from the Eastern Pacific and Carribean (Wulff et al., 2017), but our results indicate a possibility of west Pacific forcing as well.

After development in weeks 1 to 2, it is clear that the Rossby Wave comprising the teleconnection is quasi-stationary, as its lifetime extends into week 3 to 6 (Fig 5J-L). This results in negative t2m anomalies in west and central Europe when WPD was negative and positive t2m anomalies when WPD was positive (Fig 5M,O). In the negative phase a cool western Europe (relative to the climate normal from 1980-2021) is flanked by high pressure and high t2m in Eastern Europe and Russia (Fig. 5J,M). This cold-warm t2m dipole has been reported by earlier studies as well (Behera et al., 2013). In fact, given that the negative phase of WPD is occurring more frequently since 1990 (Fig. 4A), we expect that in recent summers this t2m dipole pattern has become more frequent, which is exactly what Lee et al. (2017) have reported.

## 6  Modulation

As above, we can look at corresponding WPD and t2m phases and study the teleconnection pathway when it is fully present. But we are also interested in cases where positive or negative WPD phases produce Pacific heating contrasts, but where the full atmospheric response is lacking and the QSRW fails to reach Europe because of a modulating process.

ENSO's influence on the large scale background circulation might be such an indirect modulator (Ding et al., 2011; Schneidereit et al., 2012; O'Reilly et al., 2019). An example is the cooling of the atmosphere in the entire tropical belt by a strong La Niña episode. This changes the zonal mean equator-to-pole gradient and therefore the waveguide (Ding et al., 2011). Modulation by ENSO can also happen in a zonally asymmetric way by strengthening the western north Pacific Monsoon, or Indian Summer Monsoon (Ding et al., 2011; Di Capua et al., 2020). The evolution of ENSO, i.e. whether it is strengthening, decaying, or persisting, is important as well (e.g. Jong et al., 2020). Strong expression of a pattern in the preceding winter can namely leave an imprint on extra-tropical SSTs, which leads to favourable or unfavourable diabatic interaction with QSRWs in subsequent seasons (Vijverberg and Coumou, 2022).

We investigate whether positive and negative phases of the summertime teleconnection occur under distinct ENSO evolutions, particularly focusing on ENSO evolution in the central to eastern Pacific (Niño 3) which is less directly related to our WPD index (Fig 3B). Plotted in Figure 6 is the monthly mean evolution of the relative Niño 3 index in each combination of phases (negative WPD phases in blue, positive WPD phases in red, with opacity denoting whether or not we see a European

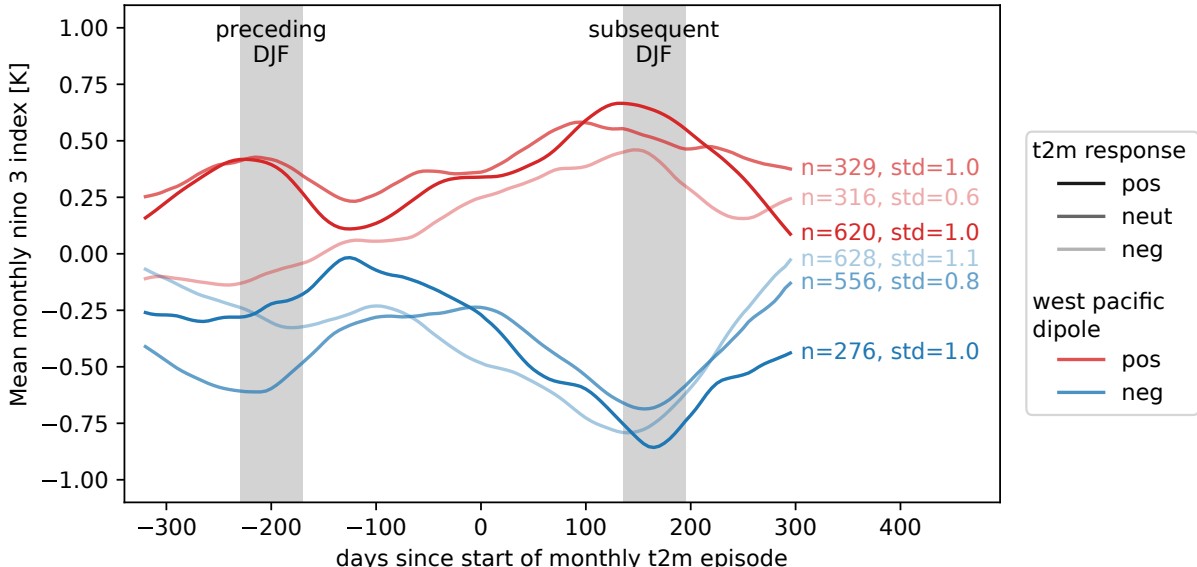

**Figure 6.** Evolution of the monthly mean relative Niño 3 index, computed per group of samples (determined in JJA). Plotted are groups with positive (red) and negative (blue) WPD phases (based on the SST anomaly in week -3 to 0, sometime in JJA). The phases tend to occur when ENSO evolves from neutral to El Niño or La Niña. Plotted with opacity is the European t2m response in week 3 to 6 (a moment in JJA that we set to zero on the x-axis). This response seems independent of the ENSO state. Samples come from the period 1980-2021. Grey boxes indicate moments that are guaranteed to fall in winter (DJF), as this depends on whether x=0 occurs early or late in summer. Number of samples and standard deviation of DJF values are annotated per group.

t2m response). As expected, positive and negative WPD phases are El-Niño-like and La-Niña-like, respectively (red lines are generally on top of blue lines, Fig. 6). This differentiation between red and blue actually increases when moving from preceding DJF, via JJA (zero on the x-axis), to subsequent DJF. This confirms that summers with a strong forcing pattern, in our case

the non-neutral WPD phases, relate to the strengthening of ENSO (Ding et al., 2011): The negative summertime WPD phase occurs during years when ENSO migrates from an average neutral state to a pronounced La Niña state. Vice versa, the positive WPD phase occurs when ENSO migrates from an average neutral state to an El Niño state. This does not imply that ENSO also fully determines the resulting t2m response: A negative t2m response is likely after a negative WPD phase (and vice versa) given the significance of the teleconnection (Fig. 4C). But within each WPD group (comparing blue lines of different opacity,

and red lines of different opacity), there are no distinct ENSO states related to the 'pos,pos' (darkest red) and 'neg,neg' (lightest blue) teleconnection occurrences (Fig. 6).

If not by ENSO, the QSRWs from west Pacific heating anomalies can also be modulated by other factors. For negative teleconnection phases little was found, but for positive WPD phases we plot the spatial composite patterns of SST and u300 (Fig. 7). The composite plots show the state in week -3 to 0, so before the QSRW occurs, and show the developing QSRW

itself with Z300 in week 1 and 2, with on the left samples leading to a lagged, positive European t2m response, and on the right

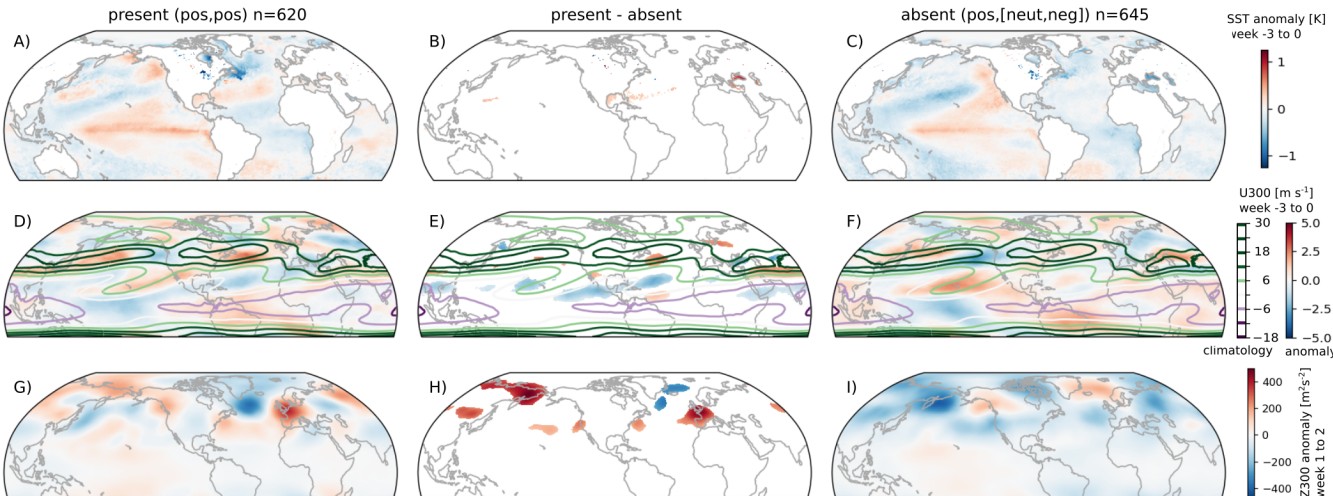

**Figure 7.** Modulation of the teleconnection when the west Pacific dipole in week -3 to 0 is in its positive phase. Left column: samples with positive t2m response in week 3 to 6, meaning a positive phase teleconnection (n = 620). Right column: samples resulting in a neutral or negative t2m response, meaning an absence of the teleconnection (n = 645). Middle column: difference between panels in the left and right column. Values found to not exceed the difference as expected by random chance are masked ($\alpha = 0.1$, more detail in Section 5). Top row: composite anomalies of SST in week -3 to 0. Middle row: composite U300 anomalies in week -3 to 0. Contour-overlay shows the summertime climatological U300 value. Bottom row: composite Z300 anomalies in week 1 to 2. Composites are extracted from the period 1980-2021 (climate normal also defined from 1980-2021).

all those that do not. The middle column shows that with a similar Pacific WPD state, SST patterns are hardly significantly different, except for a branch of warm Atlantic SSTs extending north-east from the Carribean (Fig. 7B). The warm branch is part of a meridional cold-warm-cold tripole in the north Atlantic that is associated with positive t2m responses (Fig. 7A). Diagnosed by earlier studies, this tripole pattern relates to the strength of the oceanic gyres in the north Atlantic, which are

partly driven by wind-stress on the ocean surface (Häkkinen et al., 2011). In the configuration of Fig. 7A, cyclonic circulation and cold SSTs prevail south of Greenland, while heat is transferred from ocean to atmosphere (Häkkinen et al., 2011). The tripole pattern spans multiple seasons and can occur already in late winter and early spring. From that moment onward it is known to precede the summertime pattern with a strong low pressure anomaly positioned south of Greenland and west of the British Isles (Fig. 5G) (Gastineau and Frankignoul, 2015; Ossó et al., 2020; Wolf et al., 2020; Beobide-Arsuaga et al., 2023).

Spring SST anomalies are particularly strengthened by a two-way coupling between ocean and jet stream. This happens as soon as the north Atlantic jet migrates northward with the change of seasons (Wolf et al., 2020; Ossó et al., 2020). Indeed we see that sharp meridional SST gradients are co-located with jet stream position over the Atlantic (Fig. 7A,D). Strong u300 anomalies are present in week -3 to 0, and show a stronger and narrower Atlantic jet as compared to the broader climatological mean jet (green contours, Fig. 7D). In the case without SST tripole, and without a t2m response in week 3-6, the jet is less

strong (Fig. 7F). Strong and narrow jets form better waveguides (Manola et al., 2013; White et al., 2022), and in this case

precede the succession of pressure anomalies extending from the Pacific to Atlantic (Fig. 7H). We therefore deduce that the seasonal interplay of SST tripole and Atlantic jet could modulate the sub-seasonal teleconnection by longitudinally guiding QSRWs from the west Pacific towards Europe.

## 7 Discussion and conclusion

In this study we have defined a west Pacific Dipole (WPD) index that captures the strength of a dipole SST pattern in the western Pacific. The index captures a pattern of opposing SST anomalies in the central-to-west equatorial Pacific and the Western North Pacific region, i.e. a dipole with both equatorial and meridional orientation, which was shown to relate to changing patterns of deep convection and large-scale SST changes across the Pacific. Specifically, the WPD targets those heating patterns that excite Quasi-Stationary Rossby Waves (QSRW) that potentially affect west and central Europe more than two weeks later.

Such teleconnections are known mechanisms for sub-seasonal predictability. Cross-correlations make clear that the emphasis of the WPD is different than, but still relates to, well-known Pacific modes (Fig. 3B,C). Particularly, the positive and negative WPD phases in summer coincide with the strengthening of ENSO (Fig. 6).

A prominent result of this study is that negative WPD phases have become more frequent over recent decades. In this phase, convection over the Maritime Continent and Western North Pacific is enhanced, and that over the central Equatorial Pacific

suppressed. Its increased occurrence reflects a long-term shift in the Pacific 'background state' towards stronger zonal and meridional SST gradients, as a consequence of the west Pacific warming relative to the central tropical Pacific. This west Pacific warming mode is thought to be a response to anthropogenic forcing, and to strengthen the Walker circulation (Funk and Hoell, 2015; Funk et al., 2018).

Coinciding with the long term SST changes we find that the WPD phase during weeks -3 to 0 becomes an important predictor

for the European t2m response in week 3 to 6 (Fig. 4B) (also van Straaten et al., 2023). We diagnose that this summertime Pacific to Euro-Atlantic teleconnection emerged after 1980, which agrees with other studies (O'Reilly et al., 2019; Sun et al., 2022). We should however be cautious about concluding that the teleconnection has been absent before 1980, as it can also relate to data quality, which improved with the advent of satellite observations (Hersbach et al., 2020). Nonetheless, we show that for the period of 1980 till 2021 connectivity is significant, and that the teleconnection pathway can be well understood.

Following the heating anomalies, a QSRW develops in week 1 to 2 and consists of known centres of action. Characteristic is the centre in the north-west US and that south of Greenland and west of the British Isles (Fig. 5H) (Wulff et al., 2017; O'Reilly et al., 2018; Vijverberg and Coumou, 2022). The QSRW then extends further east, with large anomalies over our west and central European target region, and anomalies with an opposing sign over eastern Europe and Russia (an opposition also known from Behera et al., 2013; Lee et al., 2017).

The emergence of a significant teleconnection in recent decades provides a lens through which recent summer circulation changes over Europe can be interpreted. This is relevant because warming trends are the consequence of multiple factors such as direct forcing by greenhouse gases and aerosols (Dong et al., 2017), but also circulation changes (Deser et al., 2016; Faranda et al., 2023). The changes diagnosed in this study are an increased occurrence of negative WPD and a reduced occurrence of

positive WPD, each with a respective QSRW response. The increased frequency of the negative WPD phase (Fig. 4A), would,
according to its corresponding QSRW, induce a warming in eastern Europe and Russia (Fig. 5M). Indeed, high pressure has
become more prevalent in this region (Lee et al., 2017; Kim and Lee, 2022; Teng et al., 2022), associated with a very strong
increase in heat waves (Rousi et al., 2022), with average summer t2m increasing more than in our Western European target
region (Teng et al., 2022).

The warming of Western European average summer t2m has been less severe, though still highly significant (Christidis
et al., 2015; Gutiérrez et al., 2021). A precise quantification of the contribution from changes in WPD to this trend is beyond
the scope of this study (and is also hindered by the methodological necessity to isolate the sub-seasonal teleconnection from
the trend by using 21-year rolling-window distributions of t2m, see section 4). But relative to the trend, two things can be said.
First, during the time that negative WPD has roughly doubled in frequency (Fig. 8A), it remained consistently related to the
cold Western European t2m tercile (Fig. 8B). This means that Western European warming is likely caused by other factors.

Second, one of these factors could be positive WPD. Although positive WPD reduced in frequency (Fig. 8C), we have seen
that its corresponding QSRW is potentially modulated by the situation in the Atlantic (section 6). A combination of SST tripole
and a strong and narrow jet in the north Atlantic allow the QSRW to reach western Europe. Prominent in this modulation are the
relatively cold SSTs south of Greenland (agreeing with Fuentes-Franco et al., 2022). Such relatively cold SSTs have become
prevalent since the 1980's (Chemke et al., 2020), just as the associated low pressure west of the British Isles (Faranda et al.,
2023). This implies that although the total count of positive WPD phases decreases, those that do occur are more likely to
generate a QSRW that reaches western Europe and results in warm t2m states there (Fig. 8D).

We suggest that the interplay between long-term Pacific changes and long-term north Atlantic changes be researched further.
Particularly because cold north Atlantic SSTs can become more prevalent if Atlantic meridional overturning circulation slows
down. Climate model experiments suggest that such slowdown might follow from further anthropogenic forcing and induce
changes in summertime Euro-Atlantic circulation (Haarsma et al., 2015; Rousi et al., 2021). Relevant for the interplay are also
long term changes that potentially affect jet stream strength, like Arctic amplification and aerosols (Coumou et al., 2018; Dong
et al., 2022).

Research into the Pacific-Atlantic interplay should probably not be conducted with numerical climate models only. Our
analysis namely emphasized features that climate models have difficulty capturing. One is the strengthening of zonal gradients
in Pacific SST, which we detected as the increase of negative WPD phases. We know that climate models are unable to
reproduce the observed strengthening, and simulate a weakening instead (Funk and Hoell, 2015; Wills et al., 2022; Seager et al.,
2022; Lee et al., 2022). Worrying is that even with prescribed SSTs, climate models fail to reproduce the associated dynamical
response: an increased prevalence of high pressure over western and eastern Europe (Boe et al., 2020). It thus appears that
climate models do not represent the detected QSRWs well. A similar failure to represent the teleconnection pathway happens
in the ECMWF model, despite being initialized with observed SSTs (van Straaten et al., 2023).

Resolving these issues is challenging because the shortcomings of climate models force us to use observations or reanalysis
products. These are of limited length, with few independent samples as a result. It is also challenging because a full theoretical
understanding of QSRWs on different space and time scales does not exist (White et al., 2022). We hope that the targeted WPD

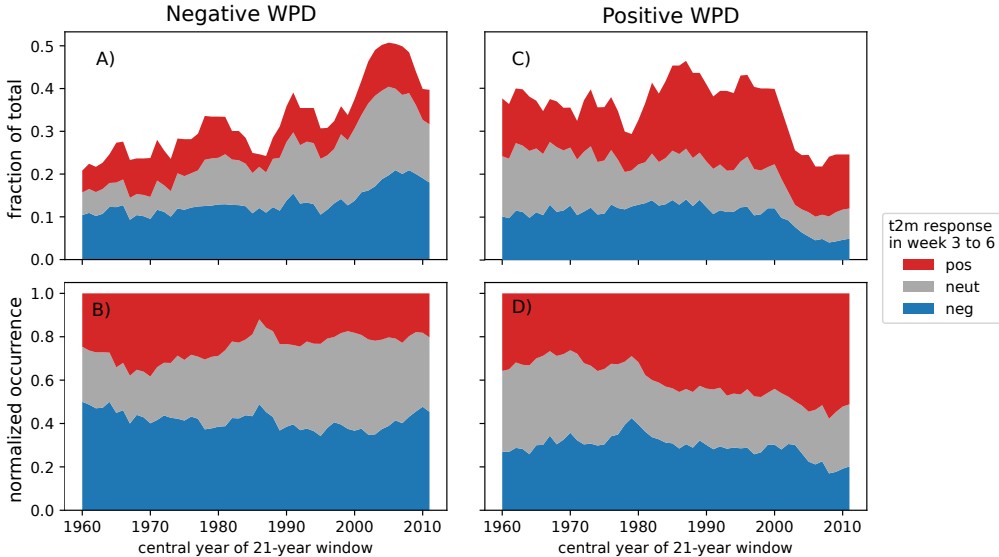

**Figure 8.** Prevalence of negative and positive West Pacific Dipole phases over time (WPD index in week -3 to 0), as colored by the resulting Western European t2m reponse in week 3 to 6. Top row: presented as fraction of the total number of samples within each 21-year rolling window. Bottom row: presented as fraction of the WPD phase itself, meaning normalized by the total prevalence of negative WPD (bottom left) and positive WPD (bottom right). Like in Fig. 4, tercile thresholds for WPD are determined over the entire dataset from 1950-2021, and tercile thresholds for t2m are re-computed in each window.

index provides future diagnostic studies with a starting point. An improved representation of the Pacific-to-Europe connection
in weather and climate models has a lot to offer. One is a better projection of European summer extremes (van Oldenborgh et al., 2022). The other is a conditional opportunity to forecast European summer circulation more than two weeks in advance (Mariotti et al., 2020; van Straaten et al., 2023).

*Code and data availability.* The ERA5 reanalysis can be obtained from the Copernicus Climate Data Store https://cds.climate.copernicus.eu. The PDO index can be accessed at https://oceanview.pfeg.noaa.gov/erddap/tabledap/cciea_OC_PDO.html. The relative ENSO indices can
be accessed at the KNMI Climate Explorer https://climexp.knmi.nl/selectindex.cgi. Python code used to conduct this study can be accessed at https://github.com/chiemvs/telegates

## Appendix A: Optimality of time and space scales

The teleconnection diagnostics presented in this study are based on a certain chosen timescales. Particularly, we relate four-week average SST anomalies (in the form of the WPD index) to four-week Western European t2m anomalies, accross a lead
time gap of two weeks. In an earlier study, this configuration of lead time and aggregation period displayed some sub-seasonal

predictability of European summertime t2m (van Straaten et al., 2022). Here we explore the position of these chosen timescales among alternatives. The black cross in Fig. A1A represents the current study, within a matrix of alternative averaging timescales (y-axis) and lead time gaps (x-axis). Panel B presents the values obtained when measuring concurrent association between WPD and western European t2m, without a lead time gap. For the concurrent, seasonal configuration (averaging timescale of roughly 12 weeks), relatively strong association values are found (top in Fig. A1B). This is the approach that most earlier studies on the summertime Pacific to Europe connection take (Ting, 1994; Ding et al., 2011; Behera et al., 2013; O'Reilly et al., 2018). However, an equally strong or even stronger association is found for the shorter averaging timescale of 8 to 4 weeks, in which case the strongest link is not found concurrently, but at a lag of zero to one week between the end of the WPD period and the start of the t2m period (Fig. A1A). That a certain separation can be optimal is even clearer when we follow the correlation values horizontally along the 2-week aggregation (Fig. A1A, second row from the bottom). These values peak at lags of 2 and 3 weeks, which corresponds to the finding that Rossby waves from tropics need time to fully establish in the mid-latitudes (e.g. Branstator, 2014). This leads us to conclude that although the Pacific to Europe teleconnection will show up in concurrent seasonal aggregates, it is quantitatively stronger in a non-concurrent framework, and thus has the character of a lagged sub-seasonal connection.

For the subset of the alternative temporal configurations highlighted by the black square in Fig. A1A, we compute correlation maps similar to those in Fig. 2B (we only investigate a subset because the computational cost is high). Fig. A3 illustrates which cells remain robustly correlated within all five of the cross-validation folds in the data from 1950-2021. The original configuration of this study is displayed in panel C with the red title. One sees that the patterns shift when alternative time scales and lead time gaps are chosen. For instance, the cluster of cells extending from component 1 along the equator to the northern coast of Papua New Guinea (e.g. Fig. A3C,D) disappears at shortest averaging time scale (panels I-K). Also influential is whether the cross-validation is performed on only the decades after 1980 (Fig. A2). In this case the equatorial extension is completely absent. Overall we see that the component 1 and 2 boxes capture those parts of the patterns that appear in the majority of the configurations.

*Author contributions.* All authors conceptualized the research and designed the experiments. CvS conducted the analysis, created visualizations and wrote the initial draft. All authors contributed through reviewing and editing.

*Competing interests.* The authors declare that they have no conflict of interest.

*Acknowledgements.* This study is part of the open research programme Aard- en Levenswetenschappen, project number ALWOP.395, which is financed by the Dutch Research Council (NWO). We thank maintainers and funders of the BAZIS cluster at VU Amsterdam for computa-

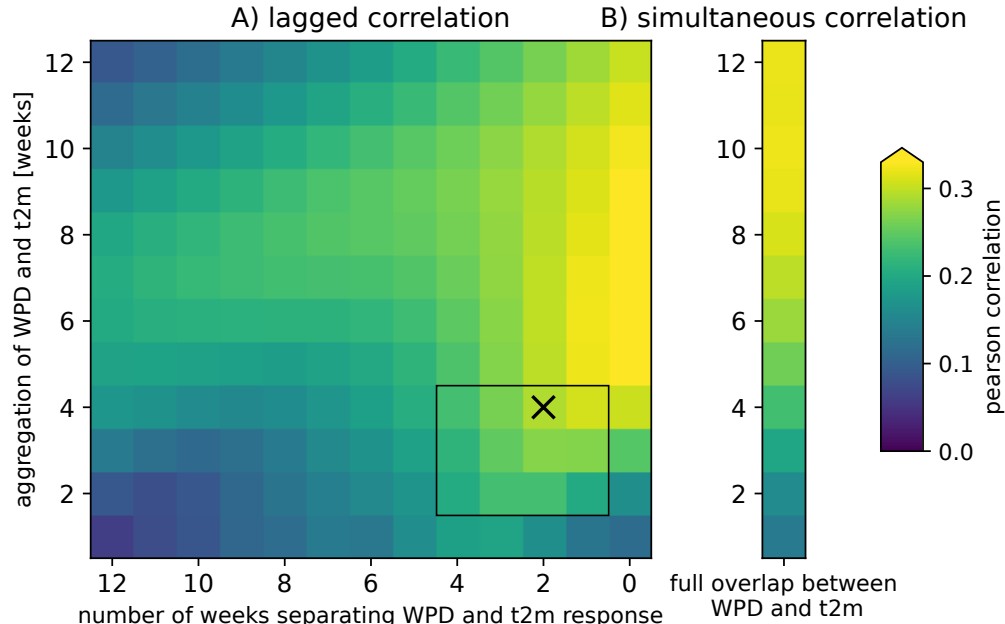

**Figure A1.** A) Pearson correlation between WPD and detrended Western European t2m at different lags and levels of temporal averaging (averaging applied equally to both WPD and t2m). Lag is defined as the number of intervening weeks (after the end of WPD and before the start of t2m response). The black cross highlights the original lag and level of temporal averaging chosen in this study. The black square highlights the combinations presented in Figs. A2 and A3. Correlation is computed using data from 1980-2021. B) Correlation obtained when WPD and t2m are fully overlapping (i.e. concurrent).

tional resources. D.C. acknowledges support from a NWO Vidi grant (Persistent Summer Extremes "PERSIST"). This research was partly funded from the European Union's Horizon 2020 research and innovation program under grant agreements 101003469 and 820970.

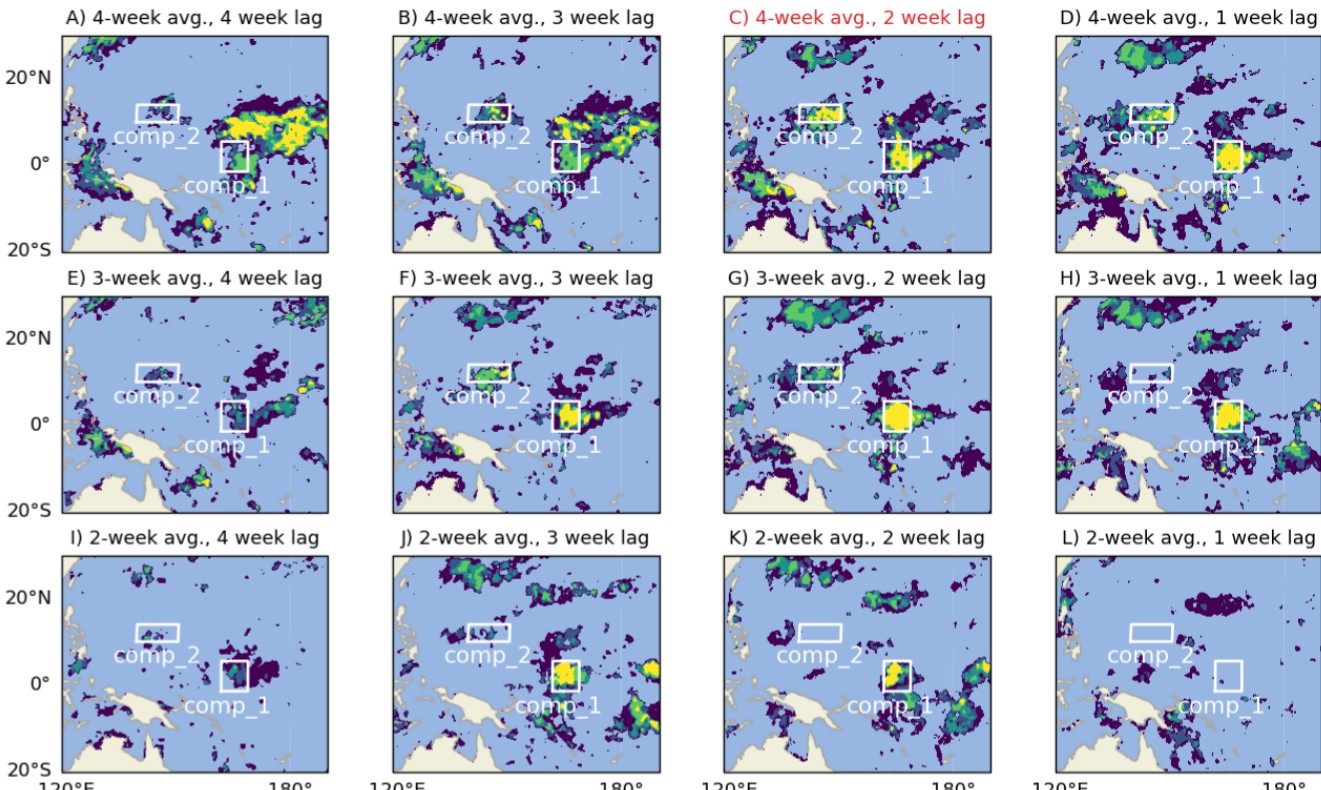

**Figure A2.** As Figure 2B, but with partial correlation computed using reanalysis data from 1980 till 2021 (5-fold cross validation). Panels illustrate different combinations of lags and time aggregation applied to the SST anomalies and west/central European t2m. Temporal aggregation (rows) is applied to WPD and t2m response equally. Lag (columns) is defined as the number of weeks after the end of WPD and before the start of the t2m response. Panel C shows the pattern under the original temporal aggregation and lag chosen in this study.

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

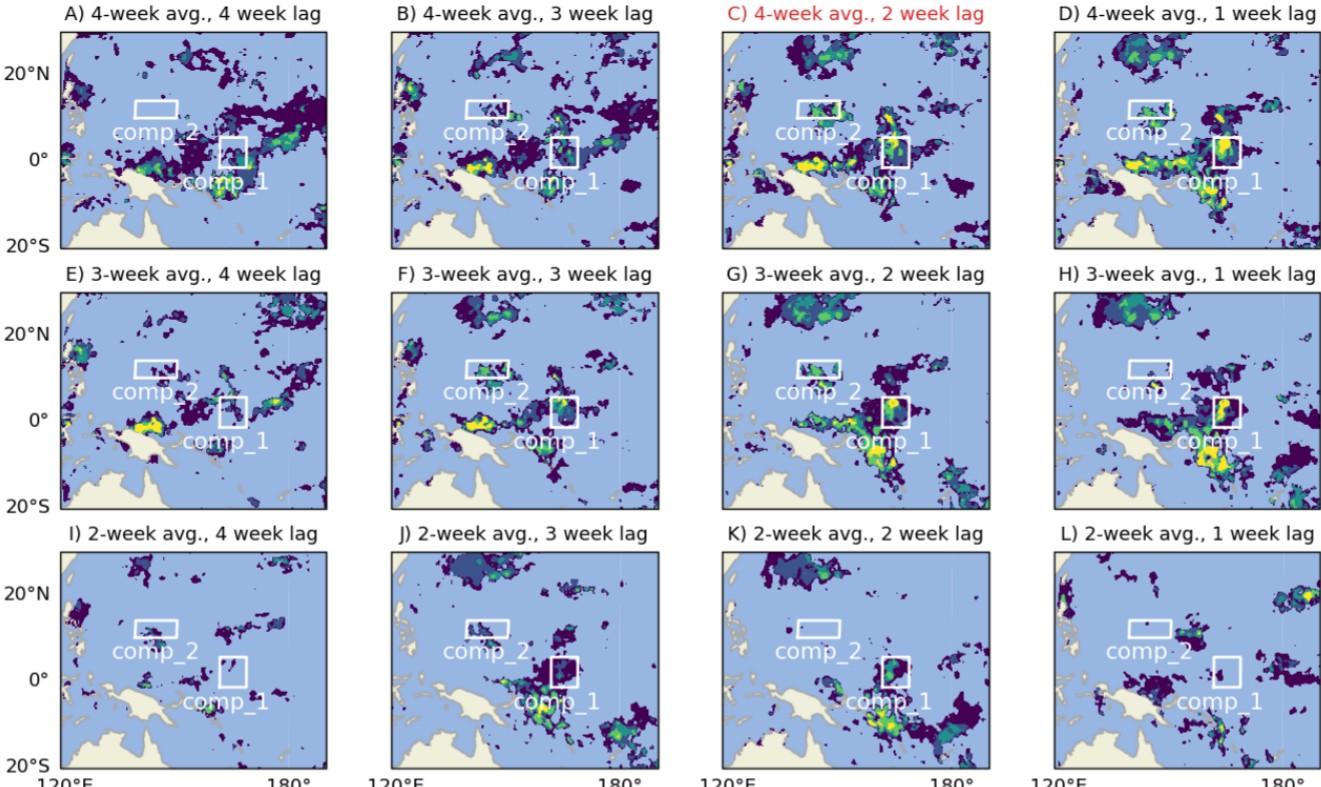

**Figure A3.** As Figure A3, but with partial correlation computed using reanalysis data from 1950 till 2021 (5-fold cross validation). Panel C shows the pattern under the original temporal aggregation and lag chosen in this study (in combination with using the full 1950-2021 data, it is therefore an exact copy of the map in Fig. 2B).

Boe, J., Terray, L., Moine, M.-P., Valcke, S., Bellucci, A., Drijfhout, S., Haarsma, R., Lohmann, K., Putrasahan, D. A., Roberts, C., et al.: Past long-term summer warming over western Europe in new generation climate models: role of large-scale atmospheric circulation, Environmental Research Letters, 15, 084 038, https://doi.org/10.1088/1748-9326/ab8a89, 2020.

Branstator, G.: Long-lived response of the midlatitude circulation and storm tracks to pulses of tropical heating, Journal of Climate, 27,

8809–8826, https://doi.org/10.1175/JCLI-D-14-00312.1, 2014.

Branstator, G. and Teng, H.: Tropospheric waveguide teleconnections and their seasonality, Journal of the Atmospheric Sciences, 74, 1513–1532, https://doi.org/10.1175/JAS-D-16-0305.1, 2017.

Buizza, R. and Leutbecher, M.: The forecast skill horizon, Quarterly Journal of the Royal Meteorological Society, 141, 3366–3382, https://doi.org/10.1002/qj.2619, 2015.

Cassou, C.: Intraseasonal interaction between the Madden–Julian oscillation and the North Atlantic Oscillation, Nature, 455, 523, https://doi.org/10.1038/nature07286, 2008.

Cassou, C., Terray, L., and Phillips, A. S.: Tropical Atlantic influence on European heat waves, Journal of climate, 18, 2805–2811, https://doi.org/10.1175/JCLI3506.1, 2005.

Chemke, R., Zanna, L., and Polvani, L. M.: Identifying a human signal in the North Atlantic warming hole, Nature communications, 11, 1–7, https://doi.org/10.1038/s41467-020-15285-x, 2020.

Christidis, N., Jones, G. S., and Stott, P. A.: Dramatically increasing chance of extremely hot summers since the 2003 European heatwave, Nature Climate Change, 5, 46–50, https://doi.org/10.1038/nclimate2468, 2015.

Coumou, D., Di Capua, G., Vavrus, S., Wang, L., and Wang, S.: The influence of Arctic amplification on mid-latitude summer circulation, Nature Communications, 9, 1–12, https://doi.org/10.1038/s41467-018-05256-8, 2018.

Deser, C., Terray, L., and Phillips, A. S.: Forced and Internal Components of Winter Air Temperature Trends over North America during the past 50 Years: Mechanisms and Implications, Journal of Climate, 29, 2237 – 2258, https://doi.org/https://doi.org/10.1175/JCLI-D-15-0304.1, 2016.

Di Capua, G., Runge, J., Donner, R. V., van den Hurk, B., Turner, A. G., Vellore, R., Krishnan, R., and Coumou, D.: Dominant patterns of interaction between the tropics and mid-latitudes in boreal summer: causal relationships and the role of timescales, Weather and Climate Dynamics, 1, 519–539, https://doi.org/10.5194/wcd-1-519-2020, 2020.

Ding, Q., Wang, B., Wallace, J. M., and Branstator, G.: Tropical–extratropical teleconnections in boreal summer: Observed interannual variability, Journal of Climate, 24, 1878–1896, https://doi.org/10.1175/2011JCLI3621.1, 2011.

Dole, R., Hoerling, M., Perlwitz, J., Eischeid, J., Pegion, P., Zhang, T., Quan, X.-W., Xu, T., and Murray, D.: Was there a basis for anticipating the 2010 Russian heat wave?, Geophysical Research Letters, 38, https://doi.org/10.1029/2010GL046582, 2011.

Dong, B., Sutton, R. T., and Shaffrey, L.: Understanding the rapid summer warming and changes in temperature extremes since the mid-1990s over Western Europe, Climate Dynamics, 48, 1537–1554, https://doi.org/10.1007/s00382-016-3158-8, 2017.

Dong, B., Sutton, R. T., Shaffrey, L., and Harvey, B.: Recent decadal weakening of the summer Eurasian westerly jet attributable to anthropogenic aerosol emissions, Nature Communications, 13, 1148, https://doi.org/10.1038/s41467-022-28816-5, 2022.

Faranda, D., Messori, G., Jezequel, A., Vrac, M., and Yiou, P.: Atmospheric circulation compounds anthropogenic warming and impacts of climate extremes in Europe, Proceedings of the National Academy of Sciences, 120, e2214525120, https://doi.org/10.1073/pnas.2214525120, 2023.

Fuentes-Franco, R. and Koenigk, T.: Identifying remote sources of interannual variability for summer precipitation over Nordic European countries tied to global teleconnection wave patterns, Tellus A: Dynamic Meteorology and Oceanography, 72, 1–15, https://doi.org/10.1080/16000870.2020.1764303, 2020.

Fuentes-Franco, R., Koenigk, T., Docquier, D., Graef, F., and Wyser, K.: Exploring the influence of the North Pacific Rossby wave sources on the variability of summer atmospheric circulation and precipitation over the Northern Hemisphere, Climate Dynamics, pp. 1–15, https://doi.org/10.1007/s00382-022-06194-4, 2022.

Funk, C., Harrison, L., Shukla, S., Pomposi, C., Galu, G., Korecha, D., Husak, G., Magadzire, T., Davenport, F., Hillbruner, C., et al.: Examining the role of unusually warm Indo-Pacific sea-surface temperatures in recent African droughts, Quarterly Journal of the Royal Meteorological Society, 144, 360–383, https://doi.org/10.1002/qj.3266, 2018.

Funk, C. C. and Hoell, A.: The leading mode of observed and CMIP5 ENSO-residual sea surface temperatures and associated changes in Indo-Pacific climate, Journal of Climate, 28, 4309–4329, https://doi.org/10.1175/JCLI-D-14-00334.1, 2015.

Gastineau, G. and Frankignoul, C.: Influence of the North Atlantic SST variability on the atmospheric circulation during the twentieth century, Journal of Climate, 28, 1396–1416, https://doi.org/10.1175/JCLI-D-14-00424.1, 2015.

Gutiérrez, J., Jones, R., Narisma, G., Alves, L., Amjad, M., Gorodetskaya, I., Grose, M., Klutse, N., Krakovska, S., Li, J., Martínez-Castro, D., Mearns, L., Mernild, S., Ngo-Duc, T., van den Hurk, B., and Yoon, J.-H.: Atlas, p. 1927–2058, Cambridge University Press, Cambridge, United Kingdom and New York, NY, USA, https://doi.org/10.1017/9781009157896.021, 2021.

Haarsma, R. J., Selten, F. M., and Drijfhout, S. S.: Decelerating Atlantic meridional overturning circulation main cause of future west European summer atmospheric circulation changes, Environmental Research Letters, 10, 094 007, https://doi.org/10.1088/1748-

9326/10/9/094007, 2015.

Hauser, M., Orth, R., and Seneviratne, S. I.: Role of soil moisture versus recent climate change for the 2010 heat wave in western Russia, Geophysical Research Letters, 43, 2819–2826, https://doi.org/10.1002/2016GL068036, 2016.

Henderson, S. A., Maloney, E. D., and Son, S.-W.: Madden–Julian oscillation Pacific teleconnections: The impact of the basic state and MJO representation in general circulation models, Journal of Climate, 30, 4567–4587, https://doi.org/10.1175/JCLI-D-16-0789.1, 2017.

Hersbach, H., Bell, B., Berrisford, P., Hirahara, S., Horányi, A., Muñoz-Sabater, J., Nicolas, J., Peubey, C., Radu, R., Schepers, D., et al.: The ERA5 global reanalysis, Quarterly Journal of the Royal Meteorological Society, 146, 1999–2049, https://doi.org/10.1002/qj.3803, 2020.

Hoskins, B. J. and Ambrizzi, T.: Rossby wave propagation on a realistic longitudinally varying flow, Journal of Atmospheric Sciences, 50, 1661–1671, https://doi.org/10.1175/1520-0469(1993)050<1661:RWPOAR>2.0.CO;2, 1993.

Hoskins, B. J. and Karoly, D. J.: The steady linear response of a spherical atmosphere to thermal and orographic forcing, Journal of the

atmospheric sciences, 38, 1179–1196, https://doi.org/10.1175/1520-0469(1981)038<1179:TSLROA>2.0.CO;2, 1981.

Huang, B., Thorne, P. W., Banzon, V. F., Boyer, T., Chepurin, G., Lawrimore, J. H., Menne, M. J., Smith, T. M., Vose, R. S., and Zhang, H.-M.: Extended reconstructed sea surface temperature, version 5 (ERSSTv5): upgrades, validations, and intercomparisons, Journal of Climate, 30, 8179–8205, https://doi.org/10.1175/JCLI-D-16-0836.1, 2017.

Häkkinen, S., Rhines, P. B., and Worthen, D. L.: Atmospheric Blocking and Atlantic Multidecadal Ocean Variability, Science, 334, 655–659,

https://doi.org/10.1126/science.1205683, 2011.

Jong, B.-T., Ting, M., Seager, R., and Anderson, W. B.: ENSO teleconnections and impacts on US summertime temperature during a multiyear La Niña life cycle, Journal of Climate, 33, 6009–6024, https://doi.org/10.1175/JCLI-D-19-0701.1, 2020.

Kim, D. W. and Lee, S.: The Role of Latent Heating Anomalies in Exciting the Summertime Eurasian Circulation Trend Pattern and High Surface Temperature, Journal of Climate, 35, 801–814, https://doi.org/10.1175/JCLI-D-21-0392.1, 2022.

Lee, M.-H., Lee, S., Song, H.-J., and Ho, C.-H.: The recent increase in the occurrence of a boreal summer teleconnection and its relationship with temperature extremes, Journal of Climate, 30, 7493–7504, https://doi.org/10.1175/JCLI-D-16-0094.1, 2017.

Lee, S., L'Heureux, M., Wittenberg, A. T., Seager, R., O'Gorman, P. A., and Johnson, N. C.: On the future zonal contrasts of equatorial Pacific climate: Perspectives from Observations, Simulations, and Theories, npj Climate and Atmospheric Science, 5, 82, https://doi.org/10.1038/s41612-022-00301-2, 2022.

Lim, Y.-K., Cullather, R. I., Nowicki, S. M., and Kim, K.-M.: Inter-relationship between subtropical Pacific sea surface temperature, Arctic sea ice concentration, and North Atlantic Oscillation in recent summers, Scientific reports, 9, 1–11, https://doi.org/10.1038/s41598-019-39896-7, 2019.

Liu, Z. and Alexander, M.: Atmospheric bridge, oceanic tunnel, and global climatic teleconnections, Reviews of Geophysics, 45, https://doi.org/10.1029/2005RG000172, 2007.

500   Ma, Q. and Franzke, C. L.: The role of transient eddies and diabatic heating in the maintenance of European heat waves: a nonlinear quasi-stationary wave perspective, Climate Dynamics, 56, 2983–3002, https://doi.org/10.1007/s00382-021-05628-9, 2021.

Manola, I., Selten, F., de Vries, H., and Hazeleger, W.: "Waveguidability" of idealized jets, Journal of Geophysical Research: Atmospheres, 118, 10–432, https://doi.org/10.1002/jgrd.50758, 2013.

Mantua, N. J., Hare, S. R., Zhang, Y., Wallace, J. M., and Francis, R. C.: A Pacific interdecadal climate oscillation with impacts on salmon production, Bulletin of the american Meteorological Society, 78, 1069–1080, https://doi.org/10.1175/1520-0477(1997)078<1069:APICOW>2.0.CO;2, 1997.

Mariotti, A., Baggett, C., Barnes, E. A., Becker, E., Butler, A., Collins, D. C., Dirmeyer, P. A., Ferranti, L., Johnson, N. C., Jones, J., et al.: Windows of Opportunity for Skillful Forecasts Subseasonal to Seasonal and Beyond, Bulletin of the American Meteorological Society, 101, E608–E625, https://doi.org/10.1175/BAMS-D-18-0326.1, 2020.

Mayer, K. and Barnes, E. A.: Subseasonal Forecasts of Opportunity Identified by an Explainable Neural Network, Geophysical Research Letters, 48, e2020GL092 092, https://doi.org/10.1029/2020GL092092, 2021.

Mayer, K. J. and Barnes, E. A.: Quantifying the Effect of Climate Change on Midlatitude Subseasonal Prediction Skill Provided by the Tropics, Geophysical Research Letters, 49, e2022GL098 663, https://doi.org/10.1029/2022GL098663, 2022.

Newman, M., Alexander, M. A., Ault, T. R., Cobb, K. M., Deser, C., Di Lorenzo, E., Mantua, N. J., Miller, A. J., Minobe, S., Nakamura, H., et al.: The Pacific decadal oscillation, revisited, Journal of Climate, 29, 4399–4427, https://doi.org/10.1175/JCLI-D-15-0508.1, 2016.

O'Reilly, C. H., Woollings, T., Zanna, L., and Weisheimer, A.: An interdecadal shift of the extratropical teleconnection from the tropical Pacific during boreal summer, Geophysical Research Letters, 46, 13 379–13 388, https://doi.org/10.1029/2019GL084079, 2019.

Ossó, A., Sutton, R., Shaffrey, L., and Dong, B.: Development, amplification and decay of Atlantic/European summer weather patterns linked to spring North Atlantic sea surface temperatures, Journal of Climate, 33, 5939–5951, https://doi.org/10.1175/JCLI-D-19-0613.1, 2020.

O'Reilly, C. H., Woollings, T., Zanna, L., and Weisheimer, A.: The Impact of Tropical Precipitation on Summertime Euro-Atlantic Circulation via a Circumglobal Wave Train, Journal of Climate, 31, 6481–6504, https://doi.org/10.1175/JCLI-D-17-0451.1, 2018.

Quinting, J. and Vitart, F.: Representation of Synoptic-Scale Rossby Wave Packets and Blocking in the S2S Prediction Project Database, Geophysical Research Letters, 46, 1070–1078, https://doi.org/10.1029/2018GL081381, 2019.

Röthlisberger, M., Frossard, L., Bosart, L. F., Keyser, D., and Martius, O.: Recurrent synoptic-scale Rossby wave patterns and their effect on the persistence of cold and hot spells, Journal of Climate, 32, 3207–3226, https://doi.org/10.1175/JCLI-D-18-0664.1, 2019.

Rousi, E., Selten, F., Rahmstorf, S., and Coumou, D.: Changes in North Atlantic atmospheric circulation in a warmer climate favor winter flooding and summer drought over Europe, Journal of Climate, 34, 2277–2295, https://doi.org/10.1175/JCLI-D-20-0311.1, 2021.

Rousi, E., Kornhuber, K., Beobide-Arsuaga, G., Luo, F., and Coumou, D.: Accelerated western European heatwave trends linked to more-persistent double jets over Eurasia, Nature communications, 13, 1–11, https://doi.org/10.1038/s41467-022-31432-y, 2022.

Sardeshmukh, P. D. and Hoskins, B. J.: The generation of global rotational flow by steady idealized tropical divergence, Journal of the Atmospheric Sciences, 45, 1228–1251, https://doi.org/10.1175/1520-0469(1988)045<1228:TGOGRF>2.0.CO;2, 1988.

Schneidereit, A., Schubert, S., Vargin, P., Lunkeit, F., Zhu, X., Peters, D. H., and Fraedrich, K.: Large-scale flow and the long-lasting blocking high over Russia: Summer 2010, Monthly Weather Review, 140, 2967–2981, https://doi.org/10.1175/MWR-D-11-00249.1, 2012.

Schubert, S., Wang, H., and Suarez, M.: Warm season subseasonal variability and climate extremes in the Northern Hemisphere: The role of stationary Rossby waves, Journal of Climate, 24, 4773–4792, https://doi.org/10.1175/JCLI-D-10-05035.1, 2011.

Schubert, S. D., Wang, H., Koster, R. D., Suarez, M. J., and Groisman, P. Y.: Northern Eurasian heat waves and droughts, Journal of Climate, 27, 3169–3207, https://doi.org/10.1175/JCLI-D-13-00360.1, 2014.

Seager, R., Henderson, N., and Cane, M.: Persistent Discrepancies between Observed and Modeled Trends in the Tropical Pacific Ocean, Journal of Climate, 35, 4571 – 4584, https://doi.org/10.1175/JCLI-D-21-0648.1, 2022.

Stan, C., Straus, D. M., Frederiksen, J. S., Lin, H., Maloney, E. D., and Schumacher, C.: Review of tropical-extratropical teleconnections on intraseasonal time scales, Reviews of Geophysics, 55, 902–937, https://doi.org/10.1002/2016RG000538, 2017.

Sun, X., Ding, Q., Wang, S.-Y. S., Topál, D., Li, Q., Castro, C., Teng, H., Luo, R., and Ding, Y.: Enhanced jet stream waviness induced by suppressed tropical Pacific convection during boreal summer, Nature Communications, 13, 1–10, https://doi.org/10.1038/s41467-022-28911-7, 2022.

Teng, H. and Branstator, G.: Amplification of waveguide teleconnections in the boreal summer, Current Climate Change Reports, 5, 421–432, https://doi.org/10.1007/s40641-019-00150-x, 2019.

Teng, H., Leung, R., Branstator, G., Lu, J., and Ding, Q.: Warming Pattern over the Northern Hemisphere Midlatitudes in Boreal Summer 1979–2020, Journal of Climate, 35, 3479–3494, https://doi.org/10.1175/JCLI-D-21-0437.1, 2022.

Ting, M.: Maintenance of northern summer stationary waves in a GCM, Journal of the atmospheric sciences, 51, 3286–3308,
https://doi.org/10.1175/1520-0469(1994)051<3286:MONSSW>2.0.CO;2, 1994.

Trenberth, K. E. and Stepaniak, D. P.: Indices of el Niño evolution, Journal of climate, 14, 1697–1701, https://doi.org/10.1175/1520-0442(2001)014<1697:LIOENO>2.0.CO;2, 2001.

Trenberth, K. E., Branstator, G. W., Karoly, D., Kumar, A., Lau, N.-C., and Ropelewski, C.: Progress during TOGA in understanding and modeling global teleconnections associated with tropical sea surface temperatures, Journal of Geophysical Research: Oceans, 103,
14 291–14 324, https://doi.org/10.1029/97JC01444, 1998.

van Oldenborgh, G. J., Hendon, H., Stockdale, T., L'Heureux, M., De Perez, E. C., Singh, R., and Van Aalst, M.: Defining El Niño indices in a warming climate, Environmental research letters, 16, 044 003, https://doi.org/10.1088/1748-9326/abe9ed, 2021.

van Oldenborgh, G. J., Wehner, M. F., Vautard, R., Otto, F. E. L., Seneviratne, S. I., Stott, P. A., Hegerl, G. C., Philip, S. Y., and Kew, S. F.: Attributing and projecting heatwaves is hard: We can do better, Earth's Future, 10, e2021EF002 271, https://doi.org/10.1029/2021EF002271,
e2021EF002271 2021EF002271, 2022.

van Straaten, C., Whan, K., Coumou, D., van den Hurk, B., and Schmeits, M.: The influence of aggregation and statistical post-processing on the subseasonal predictability of European temperatures, Quarterly Journal of the Royal Meteorological Society, 146, 2654–2670, https://doi.org/10.1002/qj.3810, 2020.

van Straaten, C., Whan, K., Coumou, D., van den Hurk, B., and Schmeits, M.: Using explainable machine learning forecasts to dis-
cover sub-seasonal drivers of high summer temperatures in western and central Europe, Monthly Weather Review, 150, 1115–1134, https://doi.org/10.1175/MWR-D-21-0201.1, 2022.

van Straaten, C., Whan, K., Coumou, D., van den Hurk, B., and Schmeits, M.: Correcting sub-seasonal forecast errors with an explainable ANN to understand misrepresented sources of predictability of European summer temperatures, Artificial Intelligence for the Earth Systems, https://doi.org/10.1175/AIES-D-22-0047.1, 2023.

Vautard, R., Cattiaux, J., Happé, T., Singh, J., Bonnet, R., Cassou, C., Coumou, D., D'Andrea, F., Faranda, D., Fischer, E., et al.: Heat extremes in Western Europe are increasing faster than simulated due to missed atmospheric circulation changes, under review, https://doi.org/10.21203/rs.3.rs-2464829/v1, 2023.

Vijverberg, S. and Coumou, D.: The role of the Pacific Decadal Oscillation and ocean-atmosphere interactions in driving US temperature predictability, npj Climate and Atmospheric Science, 5, 1–11, https://doi.org/10.1038/s41612-022-00237-7, 2022.

Vitart, F.: Madden—Julian oscillation prediction and teleconnections in the S2S database, Quarterly Journal of the Royal Meteorological Society, 143, 2210–2220, https://doi.org/10.1002/qj.3079, 2017.

Vitart, F. and Robertson, A. W.: The sub-seasonal to seasonal prediction project (S2S) and the prediction of extreme events, npj Climate and Atmospheric Science, 1, 3, https://doi.org/10.1038/s41612-018-0013-0, 2018.

Wehrli, K., Guillod, B. P., Hauser, M., Leclair, M., and Seneviratne, S. I.: Identifying key driving processes of major recent heatwaves, Journal of Geophysical Research: Atmospheres, 124, 11 746–11 765, https://doi.org/10.1029/2019JD030635, 2019.

Wheeler, M. C., Zhu, H., Sobel, A. H., Hudson, D., and Vitart, F.: Seamless precipitation prediction skill comparison between two global models, Quarterly Journal of the Royal Meteorological Society, 143, 374–383, https://doi.org/10.1002/qj.2928, 2017.

White, R. H., Kornhuber, K., Martius, O., and Wirth, V.: From Atmospheric Waves to Heatwaves: A Waveguide Perspective for Understanding and Predicting Concurrent, Persistent, and Extreme Extratropical Weather, Bulletin of the American Meteorological Society, 103, E923–E935, https://doi.org/10.1175/BAMS-D-21-0170.1, 2022.

Wills, R. C., Dong, Y., Proistosecu, C., Armour, K. C., and Battisti, D. S.: Systematic climate model biases in the large-scale patterns of recent sea-surface temperature and sea-level pressure change, Geophysical Research Letters, p. e2022GL100011, https://doi.org/10.1029/2022GL100011, 2022.

Wirth, V., Riemer, M., Chang, E. K., and Martius, O.: Rossby Wave Packets on the Midlatitude Waveguide—A Review, Monthly Weather Review, 146, 1965–2001, https://doi.org/10.1175/MWR-D-16-0483.1, 2018.

Wolf, G., Brayshaw, D. J., Klingaman, N. P., and Czaja, A.: Quasi-stationary waves and their impact on European weather and extreme events, Quarterly Journal of the Royal Meteorological Society, 144, 2431–2448, https://doi.org/10.1002/qj.3310, 2018.

Wolf, G., Czaja, A., Brayshaw, D., and Klingaman, N.: Connection between sea surface anomalies and atmospheric quasi-stationary waves, Journal of Climate, 33, 201–212, https://doi.org/10.1175/JCLI-D-18-0751.1, 2020.

Wu, Z. and Lin, H.: Interdecadal variability of the ENSO–North Atlantic Oscillation connection in boreal summer, Quarterly Journal of the Royal Meteorological Society, 138, 1668–1675, https://doi.org/10.1002/qj.1889, 2012.

Wulff, C. O., Greatbatch, R. J., Domeisen, D. I., Gollan, G., and Hansen, F.: Tropical forcing of the Summer East Atlantic pattern, Geophysical Research Letters, 44, 11–166, https://doi.org/10.1002/2017GL075493, 2017.