# Peer review of "Strengthening gradients in the tropical west Pacific connect to European summer temperatures on sub-seasonal timescales"

_Weather and Climate Dynamics, 2023_

## Author Comment (AC1)

**Final author response to reviewers for wcd-2023-6**
https://wcd.copernicus.org/preprints/wcd-2023-6/

**Title:** Strengthening gradients in the tropical west Pacific connect to European summer temperatures on sub-seasonal timescales

Summary of important proposed non-textual changes:

- Recompute results with consistent temporal aggregation for WPD and T2m. Previously this was inconsistent: (3-week sst, 2-week gap, 4-week t2m), now it is consistent (4-week sst, 2-week gap, 4-week t2m). The result is minor quantitative changes to Figures 3 to 8. Our original interpretations remain unchanged.
- Introduce a new supplementary plot (Fig. R5) for clarifying the optimality of chosen sub-seasonal timescales (4-week sst, 2-week gap, 4-week t2m), and sensitivity to this choice of timescales. The plot also highlights the difference between our lagged/subseasonal approach (panel A) and the concurrent/seasonal approach of previous studies (panel B).
- Add two supplementary plots (Fig. R7 and R8) with maps showing the robustness of correlating grid cells under different temporal aggregations. This is to help the reader understand why the current WPD boxes are placed where they are placed, and to assess alternative spatial choices.
- Addition of two panels to the original Figure 8 to visualize subdivision of negative WPD phases (Fig. R1). This eliminates one previously unexplored potential reason for the apparent contradiction in explaining the western european temperature trend. (See major comment 1 of Reviewer #1).
- Addition of significance tests to the composite plots of Figure 5 and 7. The test is based on a bootstrapping procedure, and insignificant differences are masked out (see Fig. R3 and R4). (See major comment 4 of Reviewer #1)

Summary of important proposed textual changes:

- Textually stress the separation between the timescales of phenomena. Originally we framed the long-term West Pacific Warming as 'La Nina-like'. This would suggest that it has an interannual component, but instead it is a long-term strengthening of dT/dx in the basic state which can 'amplify the SST gradients that arise due to ENSO variability, specifically the La-Nina-like anomalies'. With this textual rephrasing the descriptions become truer to previous findings (see major comment 3 of Reviewer #1).
- Clarification that the t2m response classes in Western Europe are defined as terciles of a rolling 21-year t2m distribution. This means the measured connectivity is relative to the long term warming trend that is itself the consequence of forced/unforced dynamic/thermodynamic changes. This concerns the beginning of section 4, end of section 5, and discussion. (see also major comment 1 of Reviewer #1)

**REVIEWER 1**

This work examines the process that anomalous heating over the tropical-subtropical western North Pacific can affect European temperature variability, with the modulation from the North Atlantic SST, on subseasonal timescales in the summer season. The authors defined a West Pacific Dipole (WPD) index to characterize the combined zonal and meridional heating contrast in the western North Pacific. They found that positive WPD can trigger teleconnections towards Europe, leading to warm T2m anomalies over western Europe in 3-6 weeks. They also found that the negative phase of WPD has become more frequent since 1980, consistent with the "Western Pacific warming mode". However, the summertime temperature over western Europe has not shown any cooling trend since 1980. The authors argued that the North Atlantic SST also plays an important role in modulating the teleconnections.

The results in this manuscript potentially present valuable improvement to our understanding of subseasonal climate predictability and prediction. I have some major comments and clarification questions, that hopefully could help to improve the manuscript.

**Major comments:**

1. A large portion of the manuscript depends on connecting the warming SST trends in the western Pacific to the increasing occurrences of negative WPD. I am not familiar with the "Western Pacific warming mode", but my understanding is that this is for a long-term warming trend in the SST (basic-state shifts). On the other hand, the WPD phases and their teleconnections to the European T2m characterized in this manuscript are subseasonal variability.

   I guess I am confused about the connection here: one is change in the mean (shift in the distribution) and another one is change in the standard deviation (change in the shape of the distribution). For example, in L285-289, the authors stated: "though negative WPD has occurred more frequently, western Europe has not experienced a relative cooling as expected." Based on the results shown here, WPD affects the "subseasonal variability" in western European t2m. More frequent of this event is not necessarily equal to change in seasonal mean temperature, if the standard deviation has become larger (e.g., more frequent cold subseasonal events, but also more frequent and stronger warm subseasonal events).

   In short, it would be helpful if the authors clarify the different timescales across the phenomena they have mentioned in the manuscript. Also, in L290-296, the authors examined that the positive WPD phase has become more likely to result in positive t2m anomalies in western Europe. It is also helpful to show the negative WPD phase: has negative WPD phase become less likely to result in negative t2m anomalies in western Europe?

   In this comment you point to the apparent contradiction between the long-term trend towards more frequent negative WPD phases (which would result in cooler than normal western European t2m) and the obvious strong increase in summertime western European t2m. You seem to ask us three questions:

   **1 Can I interpret the long-term Pacific trend as a shift in the basic state, and is this decoupled from the teleconnection which seems to be a sub-seasonal form of variability?**
   We argue in the paper that yes, the long term trend is a shift in the basic state of the Pacific, but also that it is highly coupled to the sub-seasonal scale. Therefore the answer to the second part of your question is negative. The WPD index is simply a temperature difference between central and western Pacific on a 4-week timescale. Since the long term trend shifts the mean/basic state to stronger temperature differences, this leads to more frequent extreme (in this case negative) WPD states. The WPD states are based on a simple thresholding of the T-differences, which are now more often exceeded. See our response to your major comment 3 for how we clarify the wording.

   **2 Is such a decoupling between long-term trend and sub-seasonal variability also applicable to the western European t2m? Meaning the sub-seasonal connection from WPD to t2m only changes variability (e.g. more short cold events), whereas the mean t2m warming would be unaffected?**
   Also for the t2m response we think that such a conceptual decoupling makes little sense. The long term trend in t2m is affected by many factors, among which are forced thermodynamic changes and forced and unforced dynamical changes. Explaining long term trends would require an investigation of all those factors, and is not a goal of this paper. To make this clearer we propose to remove the last sentence of the abstract, and to rewrite/de-emphasize a few other sentences.

   Our actual goal is to show that a summertime teleconnection with sub-seasonal character seems to have emerged amid long-term changes. To this end our analysis is based on a forced decomposition. As will be explained in response to major comment 2, we implicitly detrend the response in western Europe by defining the cold and hot t2m classes within a rolling 21-

year window. In the manuscript we mention that without such a procedure any diagnostic measure of sub-seasonal connectivity in the data would be muddied by the thermodynamic contribution to the long term t2m trend. In short: the sub-seasonal connection, as we define it, cannot contradict the long term trend because it is defined relative to it. There only appears to be a contradiction because of carelessly worded sentences such as the one you quote:
*"though negative WPD has occurred more frequently, western Europe has not experienced a relative cooling as expected."*
We propose to clarify such phrasing as:
*"as negative WPD has occurred more frequently, and significantly relates to the cold tercile of the rolling 21-year temperature distribution, it seems that without this teleconnection the warming due to other factors in this region could have been even stronger.*
With this last statement we acknowledge that the teleconnection (though defined in a rolling fashion) could be one of the factors influencing the long term trend. Additionally we want to more clearly state in the beginning of section 4, the end of section 5, and in the discussion, that the boundaries of the t2m classes shift due to the combined effect of other factors.

**3 Does the apparent contradiction occur because negative WPD phases (though more frequent) become less good at causing negative t2m responses.**

This is an interesting suggestion. After all, in Figure 8 we present an analysis that shows that over time positive WPD phases result in more positive t2m anomalies, possibly due to the influence of the Atlantic jet. Why could the converse not also be an explanation for the heating in western Europe, namely that the negative WPD phase has become less effective in resulting in negative t2m anomalies? We propose to add this decomposition of t2m responses as extra panels in the bottom row of Figure 8:

[Figure]

Fig. R1, New proposed panels for Figure 8, now including negative WPD phases in the top row.

In it one can see that the blue fraction in the top right panel is roughly constant over time. Negative WPD is thus as likely to be followed by a relatively cold t2m episode in the beginning of the data, as at the end of the data. Therefore the answer to your question is no. We propose to incorporate this in the part of the discussion where Figure 8 is discussed.

2. Also relevant to the timescale issue. In Fig. 4, WPD index is not detrended, but t2m over western Europe is detrended. As the main purpose is to examine how WPD affects western European t2m on subseasonal timescales, why the inconsistency here?

First, we don't think this is inconsistent, but actually in line with the respective properties of the WPD precursor and t2m response. For European t2m we look at mean temperatures in a particular region and this mean has a strong upward trend. For WPD classification we look at a dipole and thus a difference in temperature between two regions and this is far less affected

by global mean long-term warming trends.

Second, as said above, we need to implicitly detrend the t2m response because otherwise the (pos,pos) class would be artificially overpopulated in the last decades by the thermodynamic trend, and any measure of sub-seasonal connectivity would be distorted.

Third, we could treat precursor (WPD) and response (t2m) more consistently by detrending the WPD index as well. But Figure 3A already provides a hint that this will not alter results much because WPD is hardly trended. To demonstrate this we present here a version of Fig. 4 with detrended WPD (Fig. R2). Results are not greatly altered as the increase of negative WPD phases in recent decades is still present, just like the emergence of the teleconnection (red line Fig R2). There is more emphasis on positive WPD phases in the last 40 years. But the overall change remains small, and as said, we think our original approach is in line with the properties of the data. Therefore we refrain from detrending WPD.

[Figure]

Fig. R2 Version of Figure 4 achieved when detrending WPD. Will not be adopted in the paper.

3. The "Western Pacific warming mode" frequently mentioned in this manuscript (Funk and Hoell, 2015) characterizes the warming extending from the subtropical North Pacific to the subtropical South Pacific. On the other hand, the "La Nina-like warming" has been commonly referred to the trend in the zonal SST gradient in the equatorial Pacific, especially the cooling over the equatorial eastern Pacific (e.g., Seager et al. 2022; Lee et al. 2022). It would be better to distinguish these two. I don't think the "Western Pacific warming mode", "La Nina-like warming" or the cooling over the equatorial eastern Pacific are interchangeable. (e.g., L146-149).

In terms of dominant timescales the warming mode and ENSO-variability are indeed not interchangeable (in fact, the West Pacific Warming Mode is diagnosed by Funk and Hoell (2015) in ENSO-residual sst's, implying at least some first-order independence). We agree that ENSO is a shorter term (inter-annual) variability than the decadal warming. The long term warming results in a basic state with a stronger SST gradient over the whole pacific basin, because warming is more dominant in the west pacific. This increase in overall gradient aligns with the slight negative trend in the ENSO indices, which, when viewed spatially, is most expressed in the relative cooling of the equatorial eastern Pacific. Despite the difference in timescales and different spatial emphasis the two phenomena are thus connected by a similar effect on the overall gradient: the long term warming creates a pattern that just like the ENSO trend, and especially during La-Nina episodes, leads to strong gradients over the entire pacific.

Funk and Hoell (2015) describe the long term warming as resulting in: "a stronger western Pacific SST gradient which can interact with La Niña to intensify the Walker circulation and modulate Northern Hemisphere upper circulation patterns"

To be more faithful to this view (and to not make the phenomena appear interchangeable) we

now refer to west pacific warming as 'amplifying La Nina-like gradients' instead of 'La Nina-like'.

4. Statistical significance tests should be included in the composite plots (Figs. 5 and 7). Also, whether the differences in Figs. 5 and 7 are statistically significant should be addressed. This is a valid point, and is also raised in Major comment 3 by Reviewer #3. To include statistical significance tests we decided to adopt a block-bootstrapping procedure. The composite means in Fig 5 and 7 are based on groups delineated by the WPD and t2m tercile classes. These groups can be considered subsets of the entire population of summertime anomalies from 1980-2021. Our first step was to resample that entire population of anomalies to obtain 500 draws of potential composite means when group sizes are as in Fig 3C. The second step was to subtract those composite means to exactly re-create the situation in the middle columns of Fig. 5 and 7. The result is a sampling (null) distribution of differences that could arise by chance. Then we perform a two-sided test per gridcell, which tests whether the actual difference falls outside the empirical quantiles of the sampling distribution (q0.025 and 0.975 for alpha = 0.05, or q0.05 and 0.95 for alpha = 0.1). For grid cells where the test fails, the difference is insignificant and therefore masked. This procedure will of course be described in text and caption, and the proposed new panels for Fig. 5 and 7 are in Figs R3 and R4.

[Figure]

Fig. R3 proposed new panels for Fig. 5. Middle column only displays significant differences (conservative block bootstrapping, with alpha = 0.05). Also the t2m composite is extended to include eastern Europe and Russia.

We chose block-bootstrapping because of the potential temporal overlap and dependence between samples in our population. With block bootstrapping one randomly samples blocks of samples for which dependence can be expected. In this case we employ entire summer seasons as block size, which can be considered conservative because the actual groups need not contain a summer in its entirety. The result is that the null distribution might display a heightened variability, making it more difficult to pass the test. Another reason why the null distribution can be considered conservative is because it contains differences that might have arisen by more than chance alone. The samples are namely taken from the entire 40 years and can therefore contain diluted versions of the studied effect. Overall we can thus be confident that remaining (unmasked) differences are highly significant.

5. The teleconnection patterns shown in Figure 5 is quite similar to the circumglobal teleconnection pattern shown in Ding et al. (2011). As mentioned in L194, Ding et al. (2011) also showed similar heating dipole pattern. Furthermore, Ding et al. (2011) also suggested this pattern is more frequent in the developing ENSO summer (Fig. 11a in Ding et al. 2011). I

am wondering how much the patterns shown in this manuscript are similar to the CGT pattern in Ding et al. (2011). Is it possible there are actually the same thing?

They indeed seem to be the same phenomena. In both studies, the teleconnection mechanism is one of the tropics influencing the midlatitudes through Rossby waves. One further similarity is indeed the preferential occurrence in summers preceding ENSO peaks. However, we think we also shed a different light on the phenomena. We show that the pattern that shows up in concurrent seasonal analyses, as in the analysis of Ding et al. (2011), has a sub-seasonal substructure consisting of a lagged relation, as the wave pattern generally takes some time to establish. In the analysis of Ding et al. this timing aspect is masked by the fact that they take concurrent JJA averages. In this sense our approach that takes a lag of 2 weeks is closer to the physical reality of the time needed for such a connection (at least in numerical experiments (Branstator, 2014)). Quantitatively the link also seems somewhat stronger than concurrent analyses. We propose to provide evidence for this in the supplementary material. See minor comment Lines 69-71 of Reviewer #2, and major comment 2 of Reviewer #3.

6. Throughout the manuscript (especially in Section 5), the authors mentioned other studies frequently, e.g., Ding et al. (2011), Vijverberg and Coumou (2022), SEA pattern, Behera et al. 2013, O'Reilly et al. (2018) etc. The authors should provide more background information regarding these studies. Also, there are lots of consistency between your work and previous results. Which part of their results support the element you mentioned here? What are the unique parts of your results? We should not assume all the readers have read all these references or will read all of them along the way.

You are correct. In section 5 we use the composites to discuss the wave pattern, and we highlight the centres of action that match the patterns found in these other studies. We generally do not discuss differences (except for e.g. the difference with the concurrent seasonal analysis of Ding et al. (2011)). We propose to expand the text in section 5 to also cover differences.

**Minor comments:**

L94-97: Please include the longitudinal ranges of Nino3 and Nino4 regions. Also, how about the climatology? Or SST anomalies are relative to zonal mean (between 20N-20S), rather than relative to any climatology?

Ranges are now included. And we expand the sentence:

*"Relative ENSO indices are computed like regular ENSO indices as the average of the SST anomalies within a Niño region (climate normals defined from 1981-2010), from which the average SST anomaly between 20N and 20S is subtracted. This makes the indices less distorted by global warming and more useful for describing teleconnections (van Oldenborgh et al., 2021)."*

L122: "eastern" edge of the Nino4 area: "western"

replaced

L124-125: please provide the lon/lat ranges of the components 1 and 2.

Now included

L194: "Both types of heating contrasts were found to be important by Ding et al. (2011)" -> important to what?

This should be "for the large-scale circulation (z200) over Europe". We will include this in text.

L208-209 & L283: why not include t2m anomalies over Eastern Europe and Russia in Figure 5?

This was not included because we did not download that data. But now this is included in the renewed Figure 5. See Fig R3..

L250-254 & Fig. 7: it would be helpful to also include wave patterns (Z300 anomalies) in Figure 7.

We agree and propose to include panels with Z300 as in Fig. R4. In Panels G and I one sees that up to the west coast of North America the response to positive WPD is similar (in both the bottom left and

bottom right, high pressure is centered over the Alaska/Vancouver area), but that the significant difference (panel H) exists further east, with the low pressure south of greenland and the high pressure over europe. This supports our conclusion that the strength of the Atlantic jet modulates the downstream travel of the Rossby wave response.

[Figure]

Fig. R4. Proposed new panels for Figure 7. Bottom row now displays z300 anomalies. Middle column shows only the significant differences (conservative block bootstrapping, as per major comment 4).

L281-284: Are there any evidences supporting that the more frequent warming & high pressure over Eastern Europe and Russia are statistically significant related to the increasing WPD negative phase? This evidence provided in this study is indirect, basically because the reasoning consists of a combination of Figure 4 and Figure 5. Figure 4 provides evidence that the link to western europe has become statistically significant. Figure 5 shows that the surface imprint of the teleconnection pattern is opposite for Western Europe and Eastern Europe and Russia. This is now clearer because we've added the t2m in that region to the composite plots, as per minor comment L208-209. We however consider it out of scope for the current manuscript to provide direct evidence by repeating the analysis of statistical significance behind Figure 4, but with a different, Russian response variable.

**REVIEWER 2**

The study by van Straaten et al investigates the link between sub-seasonal variability in tropical Pacific convection, closely related to ENSO, and summertime circulation over the Euro-Atlantic. The study shows that SST gradients associated with the dipole represent a combination of ENSO variability and West Pacific warming. These gradients in the West Pacific are followed by quasi-stationary waves that linger for multiple weeks. Situations with La Niña-like gradients are followed by high-pressure centers over Eastern Europe and Russia, three to six weeks later. The study also confirms earlier findings that the summertime connectivity between the Pacific and Europe has shifted in recent decades and partly explains the increased occurrence of high sea level pressures and summer temperatures over the European continent. The evidence the authors show is compelling and extends to sub-seasonal scales the findings reported by other authors who have analysed teleconnections from the Pacific towards Europe. I think the study is worth publishing after minor amendments to the text and figures.

Minor comments:

Line 44: There is a typo in consequence.
Fixed

Lines 69-71 Even when a two weeks lag has been used, I consider it important to show that this is the optimal time lag since it is not so obvious that the signals arrive with the same lag.
This links to the question of whether this is a sub-seasonal or seasonal/concurrent phenomena (major comment 5 of Reviewer #1).
We agree that this is indeed good to show. We propose to do so with Supplementary plot R5. We see

that strong relations are found when looking at concurrent near-seasonal averages (9-12 weeks, panel B). But equally strong or even stronger relations are also found for shorter timescales of 8 to 4 weeks (panel A), in which case the strongest link is not found concurrently, but exists when the variables are related with a separation of zero to one week (between the end of the WPD period and the start of the t2m period). That a certain separation is required is even clearer when we follow the correlation values horizontally along the 2-week aggregation (second row from the bottom). These values peak at lags of 2 and 3 weeks, which corresponds to the common finding in literature that Rossby waves from tropics need two weeks to fully establish in the mid-latitudes (see also Major comment 2 of Reviewer #3). Overall this leads us to conclude that the link we are studying in this manuscript could be found in concurrent seasonal aggregates, but that this signal consists of an underlying lagged sub-seasonal connection. And with the hindsight of this new figure we can conclude that the timescales, as informed by our previous study, and as used in the presented analysis (4-week aggregates, and a 2-week separation) are at least in the vicinity of the optimum.
We propose to discuss this in text in the paragraph on data and t2m response.

[Figure]

Fig R5 correlation with detrended t2m at different lags and levels of temporal aggregation. Separation is defined as the number of intervening weeks (after the end of WPD, before the start of t2m response). B) is correlation when fully overlapping.

Line 86-87: A clear reason should be stated on why the different times for averaging the different variables are used. The authors provide citations to other works, but still in my view, for clarity's sake, the authors should mention in this paper the reason for using different time lengths for averaging.
The reason for the differing timescales was the fact that we used precomputed monthly indices for relative ENSO and PDO. However, we have now recomputed all results using a 4-week ('monthly') aggregation for our manually defined SST indices (WPD and WNP), which matches the timescales of ENSO and PDO, and also that of the average t2m response in week 3 to 6.

Figure 4B. It would be better to show the result in frequency, meaning the occurrences are divided by the total amount of occurrences and not occurrences.
Good idea. Though at least for panel C we do prefer to keep counts, as this gives an idea how many samples are present. These numbers will later match the number of samples going into each composite, presented in Figure 5 and 7. The proposed new plot is Fig R6. We propose to also use the fraction of total amount in the original Fig. 8. See the new Fig R1.

[Figure]

Fig R6. New version of Fig 4 with frequencies instead of absolute counts for panels A and B, and with consistent temporal aggregations (4-week wpd, 2-week gap, 4-week t2m).

**REVIEWER 3**

The paper by Van Straaten et al. studies the connectivity between the western Pacific and Western Europe, with a focus on the sub-seasonal predictability of summer European temperatures related to tropical Pacific sea surface temperatures (SSTs) as well as the modulation of the teleconnection by north Atlantic SSTs. The authors base their analysis on the ERA5 reanalysis and other observed sea surface temperature datasets.

I have a few comments that need to be addressed before a possible publication of the paper in Weather and Climate Dynamics.

**Major Comments:**

1. My first comment is that the study is based on one reanalysis only (ERA5) while other reanalyses do exist and could be used to assess the sensitivity of the results to the choice of ERA5. Given that they use the 1950-2021 period for ERA5, I am also assuming that the authors have used the preliminary ERA5 back-extension (1950-1978) and not the final ERA5 recent release since 1940. If it is not too cumbersome, I would suggest to update the analysis with the new ERA5 data (at least for the figures using years before 1979).
The study is indeed based on the preliminary back-extension of ERA5. According to the documentation https://confluence.ecmwf.int/display/CKB/The+family+of+ERA5+datasets the preliminary back-extension could suffer from too intense tropical cyclones. As we do not study these, we think that there is no reason for concern, and that differences will be negligible. Also, the redownloading of all data is quite cumbersome on the IT infrastructure that we employ, which is why we prefer not to extend to 1940.
We acknowledge that other reanalyses do exist and that these could be used to check the sensitivity to the data product. For aspects of this study this has already been done, such as for the west-pacific warming mode (Funk and Hoell, 2015), and the tropical central pacific cooling (Seager et al, 2022), meaning we can be confident that the increase of strong 4-week gradients as captured by negative WPD is robust. Nonetheless we acknowledge the limitation and propose to mention this in section 7.

2. In addition to spatial aggregation for the predictand, the authors use a monthly (4 weeks) aggregation for t2m and 3 weeks for SSTs. It would be interesting to see the results with a shorter temporal aggregation (10-15 days for instance, as the authors explicitly mentioned this period in their introduction as the time needed for the waves to establish) and comment on the potential differences. If these long averaging periods are required to obtain significant results (albeit with low amplitude for the correlations, 0.15–0.2), it is interesting information.

We agree that the sensitivity to temporal aggregation is interesting to test. See our response to the suggestion of Reviewer #2 (Line 86-87). In that response we propose to include Figure R5 in the supplementary materials. This figure displays that a two-week aggregation actually leads to less prominent correlation, but that at this two-week aggregation it is even clearer that one needs to employ a sub-seasonal (2-week) separation between SST and t2m response to find the clearest connection. This aligns well with the period needed to establish mid-latitude Rossby wave responses, as mentioned in the introduction of the manuscript.

The choice of timescales is now more elaborately discussed in the data and t2m section.

3. Statistical significance: no statistical significance of the composites is provided for figures 5 and 7. They need to be performed with an appropriate method (for a cautionary note about t-test, see "The Use of t values in Composite Analyses" by Brown, T.J., and Hall, B.L., Journal of climate, vol. 12. 2941-2944, 1999). It would be nice to have more details in the paper about the method used to derive statistical significance for Figure2 (without having to go to a previous publication). In relation to the false discovery rate procedure, it is not clear to me what alpha (page 6, line 119) is (in Wilks BAMS2016, alpha is the nominal level of the statistical test).

Significance testing has been added to composite plots. As an appropriate and conservative method we chose to do block-bootstrapping. See our response to major comment 4 of Reviewer #1

Regarding the false discovery rate procedure for the correlation plots we will add more detail to the relevant paragraph.

4. Figure 2 and the definition of the WPD index: as it is now, there seems to be a lot of subjectivity in the choice of the two boxes (comp_1 and comp_2). For instance, one could have extended the comp_1 westward and southward (including regions with significant correlations). Similarly, the comp_2 box could have been slightly extended eastward and southward. Is it possible to present a rationale for the choice of the boxes? Or at least, one would like to see a sensitivity analysis regarding the definition of the WPD index. It is a bit surprising to notice that a large spatial aggregation is performed for the predictand (t2m in Europe), contrasting with the small spatial extent of the regions used to estimate the WPD index.

Figure 2 gives the impression that the boxes could indeed have been extended to include some more significantly correlated cells. However, whether these extra cells outside of the boxes show up, depends on the exact chosen timescales and lags. In Fig. R7 we illustrate how the correlation patterns can shift. Influential is also whether one performs the stratified cross-validated correlations on the whole reanalysis period (shown) or only on the last few decades (Fig. R8). We made a subjective choice for boxes that seemed to capture those cells that are relatively consistent across aggregation timescales, lags and subsets of the data, i.e. we chose the boxes such that they would capture 'the robust centres of the phenomenon'. In a subjective sense this means that the presented analysis is one of the more robust options among all possible placements of the boxes. Objectively it is a fact that the SSTs in these boxes were the most useful predictor in a previous study (van Straaten et al. 2023, something we already mention in the manuscript). And it is also a fact that in the current diagnostic framework the placement works, because links to Europe are found that are in agreement with previous studies. Still we want to acknowledge that alternatives could potentially also work. We propose to include Fig R7 and R8 as supplementary material to illustrate how correlation patterns can shift when different choices are made. In this way the reader can form an opinion about alternative definitions of the index. In the text of section 3 where the boxes are introduced, we propose to discuss the influence of aggregation and lag on potential placement, and to refer to the supplementary material.

[Figure]

Figure R7 As Figure 2B, with partial correlation computed on reanalysis data from 1950 till 2021 (5-fold cross validation). Panels illustrate different combinations of lags and time aggregations to assess alternatives for box placement. The results of this study are based on the aggregation and lag highlighted in red.

[Figure]

Figure R8 As Figure R6, but with partial correlation computed on data from 1980 till 2021 (5-fold cross validation). The results of this study are based on the aggregation and lag highlighted in red.

**Minor comments**:

- Page 2, line 34: Many studies suggest that the contribution of ocean SSTs (including therefore La Niña) to the Russian heatwave is very weak (see Dole et al., 2011; Hauser et al. 2016; Wehrli et al. 2019).
  This is indeed an interesting contrast with the cited study of Schneidereit et al. (2012). Perhaps one way to understand the conundrum is that these three are modeling studies, which, as we also highlight in the introduction, can have problems simulating this type of teleconnection. The study of Schneidereit on the other hand uses reanalysis. Also, the small contribution in the case of Wehrli et al. (2019) might be a consequence of the experimental protocol, which derives SST's influence by comparing to a small collection of reference states that are neighboring in time and perhaps not so different. But nonetheless, we acknowledge that the issue is not settled. We propose to cite the mentioned studies, and state that the influence of the La Nina on the russian heatwave is debated.
- Page 2, line 51: this is a reference to a paper under review, I am not sure about the WCD policy about that.
  The paper is now accepted.
- Page 4, line 100: using this PDO definition where global mean SSTs are removed might not be the best option This assumes that any anthropogenic influence on PDO behavior is fully removed by subtracting the time evolving changes in global mean SSTs from local SST changes in the PDO region. This is unlikely as it is known that there are different forced warming rates between the PDO region SSTs and the global mean SSTs. This means that an anthropogenic signal is likely to be aliased when using this PDO index definition (see Bonfils and Santer, 2011).
  Thank you for this insight. This definition was employed by the institute from which we obtained the data: https://oceanview.pfeg.noaa.gov/erddap/tabledap/UW-JISAO_PDO.html
  But we notice now that the updating of this data is also discontinued. So instead we propose to replace it with a PDO index that does not suffer from this issue https://oceanview.pfeg.noaa.gov/erddap/tabledap/cciea_OC_PDO.html and update the data description accordingly. Replacing the index hardly changes the presented correlation coefficients in Fig. 3, and PDO remains uncorrelated to the response.
- Page 4: there is no reference to figure 1A, perhaps add it line 109.
  Good suggestion
- Page 6, lines 126–129: I think that this statement (that the WPD captures the combined heating contrast …) need a bit more details, or maybe even an additional figure. It is written as if it is an obvious fact, but it is not.
  True, we propose to amend the text.
- Page 6, figure 3 caption: why do you use 4 weeks instead of 3 for PDO and Niño indices?
  This is now harmonized, all variables are at the 4 weeks/monthly timescale. See also comment Line 86-87 of Reviewer #2.
- Page 6, lines 131–134: I do not understand the logic here. The WPD index is based on two boxes that are located in the Western Pacific, and one of the boxes is outside the equatorial region. It is not clear to see how this can be related to the equatorial SST gradient across the whole Pacific basin. Please explain.
  True, the index as defined is not purely based on equatorial gradients, but it is a combination of equatorial and meridional. As mentioned in the manuscript both types were found to be important by Ding et al. (2011). We propose to amend this text to make it clearer.
- Page 7, lines 155–159: why not using the "detrending" approach outlined for Figure 2 instead of using a 21-yr rolling window?
  The procedure with the rolling 21-yr window is more involved than detrending with linear regression (the approach taken for Figure 2), but we think it is particularly suited for the data, because of the non-linearity in the long-term t2m warming in the last few decades. The reason we detrend with regression in Figure 2 is that we wanted to simultaneously correct for inflation due to autocorrelation. which is possible when both factors (autocorrelation, and long-term trend) are jointly estimated and removed within a single regression framework. That the approach of Figure 2 is different does not lead to inconsistencies, as after all, the grid-based correlation is only used to inform the placement of the boxes, after which WPD is

defined on SST anomalies that are only de-seasonalized. See also our reply to major comment 2 of Reviewer #1.

- Page 7, lines 163–165: please specify what are exactly the equatorial (SST?) gradients you are talking about.
  In this case this is the increase in gradient as seen in the long-term warming of the west pacific (WNP) and the cooling of the central pacific (Nino3). This could have been clearer indeed, and is also a gradient that is not fully equatorial. As above, we propose to amend this text.
- Figure 7: the contours are very difficult to see, can you enlarge the plots and make thicker contours?
  Contours are now made thicker. See Fig R4.

**References**:

Bonfils C, Santer BD (2011) Investigating the possibility of a human component in various Pacific Decadal Oscillation indices. Clim Dyn 37:1457–1468. https://doi.org/10.1007/s00382-010-0920-1

Dole, R., Hoerling, M., Perlwitz, J., Eischeid, J., Pegion, P., Zhang, T., et al. (2011).Was there a basis for anticipating the 2010 Russian heat wave? Geophysical Research Letters, 38, L06702. https://doi.org/10.1029/2010GL046582

Hauser, M., Orth, R., & Seneviratne, S. I. (2016). Role of soil moisture versus recent climate change for the 2010 heat wave in western Russia. Geophysical Research Letters, 43, 2819–2826. https://doi.org/10.1002/2016GL068036

Wehrli, K., Guillod, B. P., Hauser, M., Leclair, M., & Seneviratne, S. I. (2019). Identifying key driving processes of major recent heat waves. Journal of Geophysical Research: Atmospheres, 124, 11,746–11,765. https://doi.org/10.1029/2019JD030635

---

## Author Response (AR1)

**Response to reviewers, accompanying revisions for wcd-2023-6**
https://wcd.copernicus.org/preprints/wcd-2023-6/

**Title:** Strengthening gradients in the tropical west Pacific connect to European summer temperatures on sub-seasonal timescales

Highlighted line numbers (L00) concern the tracked changes document.

Summary of important non-textual changes:

- Recomputed results with consistent temporal aggregation for WPD and T2m. Previously this was inconsistent: (3-week sst, 2-week gap, 4-week t2m), now it is consistent (4-week sst, 2-week gap, 4-week t2m). The result is minor quantitative changes to Figures 3 to 8. Our original interpretations remain unchanged.
- Introduced a new appendix plot (Fig. R5) for clarifying the optimality of chosen sub-seasonal timescales (4-week sst, 2-week gap, 4-week t2m), and sensitivity to this choice of timescales. The plot also highlights the difference between our lagged/subseasonal approach (panel A) and the concurrent/seasonal approach of previous studies (panel B).
- Added two appendix plots (Fig. R7 and R8) with maps showing the robustness of correlating grid cells under different temporal aggregations. This is to help the reader understand why the current WPD boxes are placed where they are placed, and to assess alternative spatial choices.
- Addition of two panels to the original Figure 8 to visualize subdivision of negative WPD phases (Fig. R1). This eliminates one previously unexplored potential reason for the apparent contradiction in explaining the western European temperature trend. (See major comment 1 of Reviewer #1).
- Addition of significance tests to the composite plots of Figure 5 and 7. The test is based on a bootstrapping procedure, and insignificant differences are masked out (see Fig. R3 and R4). (See major comment 4 of Reviewer #1)

Summary of important textual changes:

- Textually stress the separation between the timescales of phenomena. Originally we framed the long-term West Pacific Warming as 'La Nina-like'. This would suggest that it has an interannual component, but instead it is a long-term strengthening of dT/dx in the basic state which can 'amplify the SST gradients that arise due to ENSO variability, specifically the La-Nina-like anomalies'. With this textual rephrasing the descriptions become more in line with previous findings (see major comment 3 of Reviewer #1).
- Clarification that the t2m response classes in Western Europe are defined as terciles of a rolling 21-year t2m distribution. This means the measured connectivity is relative to the long term warming trend that is itself the consequence of forced/unforced dynamic/thermodynamic changes. This concerns the beginning of section 4, end of section 5, and discussion. (see also major comment 1 of Reviewer #1)

**REVIEWER 1**

This work examines the process that anomalous heating over the tropical-subtropical western North Pacific can affect European temperature variability, with the modulation from the North Atlantic SST, on subseasonal timescales in the summer season. The authors defined a West Pacific Dipole (WPD) index to characterize the combined zonal and meridional heating contrast in the western North Pacific. They found that positive WPD can trigger teleconnections towards Europe, leading to warm T2m anomalies over western Europe in 3-6 weeks. They also found that the negative phase of WPD has become more frequent since 1980, consistent with the "Western Pacific warming mode". However, the summertime temperature over western Europe has not shown any cooling trend since 1980. The

authors argued that the North Atlantic SST also plays an important role in modulating the teleconnections.

The results in this manuscript potentially present valuable improvement to our understanding of subseasonal climate predictability and prediction. I have some major comments and clarification questions, that hopefully could help to improve the manuscript.
We thank the reviewer for reviewing the manuscript and making helpful suggestions.

**Major comments:**

1. A large portion of the manuscript depends on connecting the warming SST trends in the western Pacific to the increasing occurrences of negative WPD. I am not familiar with the "Western Pacific warming mode", but my understanding is that this is for a long-term warming trend in the SST (basic-state shifts). On the other hand, the WPD phases and their teleconnections to the European T2m characterized in this manuscript are subseasonal variability.

   I guess I am confused about the connection here: one is change in the mean (shift in the distribution) and another one is change in the standard deviation (change in the shape of the distribution). For example, in L285-289, the authors stated: "though negative WPD has occurred more frequently, western Europe has not experienced a relative cooling as expected." Based on the results shown here, WPD affects the "subseasonal variability" in western European t2m. More frequent of this event is not necessarily equal to change in seasonal mean temperature, if the standard deviation has become larger (e.g., more frequent cold subseasonal events, but also more frequent and stronger warm subseasonal events).

   In short, it would be helpful if the authors clarify the different timescales across the phenomena they have mentioned in the manuscript. Also, in L290-296, the authors examined that the positive WPD phase has become more likely to result in positive t2m anomalies in western Europe. It is also helpful to show the negative WPD phase: has negative WPD phase become less likely to result in negative t2m anomalies in western Europe?

   In this comment you point to the apparent contradiction between the long-term trend towards more frequent negative WPD phases (which by itself would have a cooling effect on western European t2m) and the obvious strong increase in summertime western European t2m. We respond by answering three sub-questions:

   **1 Can the long-term Pacific trend be interpreted as a shift in the basic state, and is this decoupled from the teleconnection which seems to be a sub-seasonal form of variability?**
   We argue in the paper that indeed the long term trend is a shift in the basic state of the Pacific. This shift in basic state is strongly linked to the sub-seasonal scale by changing the temperature-gradients and thereby preferred regions of convective activity. The answer to the second part of the question is thus negative. The WPD index is simply a temperature difference between central and western Pacific on a 4-week timescale. Since the long term trend shifts the mean/basic state to stronger temperature differences, this leads to more frequent extreme (in this case negative) WPD states. The WPD states are based on a simple thresholding of the T-differences, which are now more often exceeded. See our response to your major comment 3 for how we have now clarified the wording on this.

   **2 Is such a decoupling between long-term trend and sub-seasonal variability also applicable to the western European t2m? Implying that the sub-seasonal connection from WPD to t2m only changes variability (e.g. more short cold events), whereas the mean t2m warming would be unaffected?**
   Also for the t2m response we think that such a conceptual decoupling is not helpful. The long

term trend in t2m is affected by many factors, also thermodynamic and dynamical drivers unrelated to the SST dynamics in the Pacific. Explaining long term trends would require an investigation of all those factors, which is outside the scope of this paper. To make this clearer in the manuscript, we remove the last sentence of the abstract L14 and rewrite the last paragraph of the intro L74:

*In this study we characterize the west Pacific teleconnection to European summer temperature variability and evaluate whether the teleconnection relates long term changes in the Pacific to Euro-Atlantic circulation.*

This phrasing clarifies that our actual goal is to show that a summertime teleconnection with sub-seasonal character has emerged amid long-term changes. To this end our analysis is based on a forced decomposition. As will be explained in response to major comment 2, we implicitly detrend the response in western Europe by defining the cold and hot t2m classes within a rolling 21-year window. In the manuscript we mention that without such a procedure any diagnostic measure of sub-seasonal connectivity in the data would be submerged by the thermodynamic contribution to the long term t2m trend. In short: the sub-seasonal connection, as we define it, cannot contradict the long term trend because we analyze deviations from the trend. There only appears to be a contradiction because of carelessly worded sentences such as the one you quote:

*"though negative WPD has occurred more frequently, western Europe has not experienced a relative cooling as expected."*

We clarify this phrasing L352:

*"The emergence of a significant teleconnection in recent decades provides a lens through which recent summer circulation changes over Europe can be interpreted. The increased frequency of the negative WPD phase (Fig. 4A) would induce a warming in eastern Europe and Russia, according to its corresponding QSRW (Fig. 5M). Indeed, high pressure has become more prevalent in this region (Lee et al., 2017; Kim and Lee, 2022; Teng et al.,2022) associated with a very strong increase in heatwaves (Rousi et al., 2022), with average summer t2m increasing more than in our Western European target region (Teng et al., 2022). To a smaller extent also Western European t2m has been increasing. Methodologically we isolated the sub-seasonal teleconnection from this trend by using 21-year rolling-window distributions of t2m. During the time that negative WPD has roughly doubled in frequency (Fig. 8A), it remained consistently related to the cold Western European t2m tercile (Fig. 8B), relative to the warming trend. This means that if the teleconnection is influencing Western Europe through circulation changes, then its effect among all other factors, would be a dampening of the warming."*

With this last statement we acknowledge that the teleconnection could be one of the factors influencing the long term trend. Additionally we now clearly state throughout the manuscript that the thresholds delineating the t2m tercile classes are time-varying to focus on the sub-seasonal connectivity. L186:

*"Week 3-to-6 t2m on the other hand, shows a clear trend (Fig. 1A). This trend can be due to a combination of thermodynamic and dynamic processes that can be both of forced and/or unforced origin (see e.g. Deser et al., 2016; Faranda et al. 2023). Since this study focuses on the sub-seasonal connection between WPD and Euro-Atlantic circulation,we remove the long-term warming trend by computing t2m tercile thresholds per rolling time window of 21 summers. If not, the upper tercile 'positive' t2m class would mostly consist of samples from recent years."*

**3 Have negative WPD phases (though occurring more frequently) become less good at causing negative t2m responses?**

This is an interesting suggestion. After all, in Figure 8 we present an analysis that shows that over time positive WPD phases more often result in positive t2m anomalies, which possibly relates to modulation of the teleconnection by the situation over the Atlantic . We explore whether the converse is also an explanation for the heating in western Europe, namely that the negative WPD phase has become less effective to generate negative t2m anomalies. We add this decomposition of t2m responses as extra panels in the top row of Figure 8:

[Figure]

Fig. R1, New version of Figure 8. Left column: Prevalence of West Pacific Dipole phases over time (WPD index in week -3 to 0), as colored by the resulting Western European t2m reponse in week 3 to 6. Right column: Presented as fraction of the WPD phase itself, namely responses following negative WPD (B) and responses following positive WPD (D). Like in Fig. 4, tercile thresholds for WPD are determined over the entire dataset from 1950-2021, and tercile thresholds for t2m are re-computed in each window

One can see that the blue fraction in panel B is roughly constant over time. Negative WPD followed by a relatively cold t2m episode is thus as likely in the beginning of the data as at the end. We incorporated this finding in the clarified phrasing also discussed above L360:
*"During the time that negative WPD has roughly doubled in frequency (fig. 8A), it remained consistently related to the cold Western European t2m tercile (Fig. 8B), relative to the warming trend."*

2.  Also relevant to the timescale issue. In Fig. 4, WPD index is not detrended, but t2m over western Europe is detrended. As the main purpose is to examine how WPD affects western European t2m on subseasonal timescales, why the inconsistency here?

    First, we don't think this is inconsistent, but actually in line with the respective properties of the WPD precursor and t2m response. For European t2m we look at mean temperatures in a particular region and this mean has a strong upward trend. We need to detrend the t2m response because otherwise the (pos,pos) class would be artificially overpopulated in the last decades by the thermodynamic trend, and any measure of sub-seasonal connectivity would be distorted.
    For WPD classification we look at a dipole and thus a difference in temperature between two regions, and the detrending of these regional temperatures does not affect the WPD values. Figure 3A already provides a hint that WPD is hardly trended. To demonstrate this we present here a version of Fig. 4 with detrended WPD (Fig. R2). Results are not greatly altered as the increase of negative WPD phases in recent decades is still present, just like the emergence of the teleconnection (red line in Fig R2). There is more emphasis on positive WPD phases in the last 40 years. But the overall change remains small, and as said, we think our original approach is in line with the properties of the data. Therefore we refrain from detrending WPD.

[Figure]

Fig. R2 Version of Figure 4 achieved when detrending WPD. Will not be adopted in the paper.

3. The "Western Pacific warming mode" frequently mentioned in this manuscript (Funk and Hoell, 2015) characterizes the warming extending from the subtropical North Pacific to the subtropical South Pacific. On the other hand, the "La Nina-like warming" has been commonly referred to the trend in the zonal SST gradient in the equatorial Pacific, especially the cooling over the equatorial eastern Pacific (e.g., Seager et al. 2022; Lee et al. 2022). It would be better to distinguish these two. I don't think the "Western Pacific warming mode", "La Nina-like warming" or the cooling over the equatorial eastern Pacific are interchangeable. (e.g., L146-149).

We agree with the reviewer that careful wording is essential and we improved our manuscript accordingly. In terms of dominant timescales the warming mode and ENSO-variability are indeed not interchangeable (in fact, the West Pacific Warming Mode is diagnosed by Funk and Hoell (2015) in ENSO-residual SSTs, i.e. the SST variability with ENSO being regressed out, thus implying at least some first-order independence).

The long term warming results in a basic state with a stronger SST gradient over the whole pacific basin, because warming is stronger in the west pacific, and there is a lack of warming in the eastern equatorial Pacific. This increase in the gradient of the background state aligns with the slight negative trend in the ENSO indices (i.e. towards La Nina). Despite the difference in timescales the two phenomena are thus connected by a similar effect on the overall gradient: the long term warming creates a pattern that just like the La-Nina episodes (and the trend towards La Nina), leads to strong gradients over the entire pacific, with a warm west pacific and comparatively cool central to eastern equatorial pacific.

Funk and Hoell (2015) describe the long term warming as resulting in: "a stronger western Pacific SST gradient which can interact with La Niña to intensify the Walker circulation and modulate Northern Hemisphere upper circulation patterns"

To be more faithful to this view (and to not make the phenomena appear interchangeable) we now refer to west pacific warming as 'amplifying La Nina-like gradients' instead of 'La Nina-like'.

L44:

*"SST contrasts in both the zonal and meridional direction have increased rapidly since 1990 by what is called the 'west Pacific warming mode' (Funk and Hoell, 2015). This long term change is diagnosed in SSTs from which the first order effects of ENSO have been removed and can thus be viewed as change in the 'background state' of the Pacific. It consists of concentrated warming over the Maritime continent and the WNP region."*

Also L176:

*"These opposing trends reflect documented warming in the western Pacific, while the central to eastern tropical Pacific has not warmed (Wills et al., 2022; Seager et al., 2022; Sun et al., 2022; Lee et al., 2022). This results in a strengthening of equatorial and meridional SST gradients in the climatological 'background state' of the Pacific, which can interact with La Niña to produce a stronger Walker circulation (Funk and Hoell, 2015; Lee et al., 2022), [...]"*

4. Statistical significance tests should be included in the composite plots (Figs. 5 and 7). Also, whether the differences in Figs. 5 and 7 are statistically significant should be addressed.

This is a valid point, and is also raised in Major comment 3 by Reviewer #3. To include statistical significance tests we decided to adopt a block-bootstrapping procedure. The composite means in Fig 5 and 7 are based on groups delineated by the WPD and t2m tercile classes. These groups can be considered subsets of the entire population of summertime anomalies from 1980-2021. Our first step was to resample that entire population of anomalies to obtain 500 draws of potential composite means when group sizes are as in Fig 4C. The second step was to subtract those composite means to exactly re-create the situation in the middle columns of Fig. 5 and 7. The result is a sampling (null) distribution of differences that could arise by chance. Then we perform a two-sided test per gridcell, which tests whether the actual difference falls outside the empirical quantiles of the sampling distribution (q0.025 and 0.975 for alpha = 0.05, or q0.05 and 0.95 for alpha = 0.1). For grid cells where the test fails, the difference is insignificant and therefore masked. This procedure is described in Line 216-221, and the proposed new panels for Fig. 5 and 7 are in Figs R3 and R4.

[Figure]

Fig. R3 updated version of Fig. 5, now including eastern Europe and western Russia in t2m composites:

Composite plots illustrating the teleconnection between week -3 to 0 west Pacific SST and week 3 to 6 west-central European t2m, based on composite anomalies from the period 1980-2021 (climate normal also estimated from 1980-2021). Left column: Samples in which negative WPD phases precede the negative t2m class (n=651). Right column: Samples in which positive WPD phases precede the positive t2m class (n=631). Middle column: Difference between panels in the right and left column. Values found to not exceed the difference as expected by random chance are masked (α = 0.05, more detail in Section 5). From top to bottom: SST pattern and OLR in week -3 to 0, the subsequent z300 response in week 1 and 2, and the eventual impact on z300 and surface temperature in week 3 to 6. Black insets in panels A-F show the two components of the West Pacific Dipole index. Green inset

in panel L serves as reference for the spatial extent of surface t2m in the bottom row.

We chose block-bootstrapping because of the potential temporal overlap and dependence between samples in our population. With block bootstrapping we randomly sample blocks of samples for which dependence can be expected. In this case we employ entire summer seasons as block size, which can be considered conservative because the actual groups need not contain a summer in its entirety. The result is that the null distribution might display an elevated variability, making it more difficult to pass the test. Another reason why the null distribution can be considered conservative is because it contains differences that might have arisen by more than chance alone. The samples are namely taken from the entire 40 years and can therefore partly contain the effect defining those 40 years and whose significance is being tested. Overall we can thus be confident that remaining (unmasked) differences are highly significant.

5. The teleconnection patterns shown in Figure 5 is quite similar to the circumglobal teleconnection pattern shown in Ding et al. (2011). As mentioned in L194, Ding et al. (2011) also showed similar heating dipole pattern. Furthermore, Ding et al. (2011) also suggested this pattern is more frequent in the developing ENSO summer (Fig. 11a in Ding et al. 2011). I am wondering how much the patterns shown in this manuscript are similar to the CGT pattern in Ding et al. (2011). Is it possible there are actually the same thing?

They indeed seem to be the same phenomena. In both studies, the teleconnection mechanism is one of the tropics influencing the midlatitudes through Rossby waves. One further similarity is indeed the preferential occurrence in summers preceding ENSO peaks. However, we think we also shed a different light on the phenomena. We show that the pattern that shows up in concurrent seasonal analyses, as in the analysis of Ding et al. (2011), has a sub-seasonal substructure consisting of a lagged relation, as the wave pattern generally takes some time to establish. In the analysis of Ding et al. this timing aspect is masked by the fact that they take concurrent JJA averages. In this sense our approach that takes a lag of 2 weeks is closer to the physical reality of the time needed for such a connection (at least in numerical experiments (Branstator, 2014)). Quantitatively the link also seems somewhat stronger than in concurrent analyses. We now provide evidence for this in the Appendix. See minor comment Lines 69-71 of Reviewer #2, and major comment 2 of Reviewer #3.

6. Throughout the manuscript (especially in Section 5), the authors mentioned other studies frequently, e.g., Ding et al. (2011), Vijverberg and Coumou (2022), SEA pattern, Behera et al. 2013, O'Reilly et al. (2018) etc. The authors should provide more background information regarding these studies. Also, there are lots of consistency between your work and previous results. Which part of their results support the element you mentioned here? What are the unique parts of your results? We should not assume all the readers have read all these references or will read all of them along the way.

You are correct. In section 5 we use the composites to discuss the wave pattern, and we highlight the centres of action that match the patterns found in these other studies. We generally do not discuss differences (except for e.g. the difference with the concurrent seasonal analysis of Ding et al. (2011)). We expanded the text in section 5 to cover additional differences. These are for example qualitative differences, as highlighted by the significance testing that we now conduct. L223:

*"Visually the SST states in week -3 to 0 represent a combination of three patterns […] (iii) a PDO-like pattern consisting of the same north-east extension of warm SST anomalies, surrounded by cold anomalies to its south, west and north, which is a Pacific configuration known to provide summertime predictability for the eastern US (our Fig. 5A and Fig. 1F of Vijverberg and Coumou (2022)). Generally, patterns of SST anomalies captured by WPD resemble the results of the mentioned studies only partially. For instance, the 'cold surrounding', which is part of the PDO-like pattern, does not show up as statistically significant in this study. Such differences are understandable because the WPD index is*

*designed for direct sub-seasonal association to summertime European t2m, which, only to a lesser degree, is found in the other modes (Fig 3C)."*

With more explanation on the different methods L245:

*"This opposition matches the first mode of summertime tropical precipitation variability that was earlier found to be important for steering mid-latitude circulation (Ding et al., 2011). The main difference with the analysis of Ding et al. (2011) is that their analysis diagnosed a concurrent seasonal link, whereas our results suggest that it consists of an underlying, lagged and sub-seasonal relation (see also Appendix A)."*

As well as highlighting a methodological difference in L256:

*"These centres are part of summertime patterns diagnosed both with a focus on the US heatwaves (Vijverberg and Coumou, 2022) and with a focus on the Euro-Atlantic circulation (Wulff et al., 2017; O'Reilly et al., 2018). As the latter studies used variance decomposition methods (in contrast to our WPD- and t2m-based compositing), the found centres thus appear to be robust across methods."*

In the process of rewriting these sentences, we removed previous references to the Summer East Atlantic Pattern. We argue that it is not necessary to discuss the cyclonic anomaly south of Greenland by means of this pattern. Additionally, it could be potentially confusing, as two different definitions of SEA are present (see Osso et al. 2020.).

**Minor comments:**

L94-97: Please include the longitudinal ranges of Nino3 and Nino4 regions. Also, how about the climatology? Or SST anomalies are relative to zonal mean (between 20N-20S), rather than relative to any climatology?

Ranges are now included. And we expand the sentence: L105

*"Relative ENSO indices are computed like regular ENSO indices as the average of the SST anomalies within a Niño region (climate normals defined from 1981-2010), from which the average SST anomaly between 20N and 20S is subtracted. This makes the indices less distorted by global warming and more useful for describing teleconnections (van Oldenborgh et al., 2021)."*

L122: "eastern" edge of the Nino4 area: "western"

replaced. L143

L124-125: please provide the lon/lat ranges of the components 1 and 2.

Now included. L144

L194: "Both types of heating contrasts were found to be important by Ding et al. (2011)" -> important to what?

This should be "for steering mid-latitude circulation". (in their case z200, in our case z300). Now added to text. L41

*"Focusing for instance on positive WPD phases, the heating over the Maritime Continent and in the Western North Pacific monsoon region (north-east of the Philippines) is reduced, and the heating over the central Pacific is enhanced (Fig. 5E-F). This opposition matches the first mode of summertime tropical precipitation variability that was found earlier to be important for steering mid-latitude circulation (Ding et al., 2011)."*

L208-209 & L283: why not include t2m anomalies over Eastern Europe and Russia in Figure 5?

This was not included because we did not download that data. But now this is included in the renewed Figure 5. See Fig R3.

L250-254 & Fig. 7: it would be helpful to also include wave patterns (Z300 anomalies) in Figure 7. We agree and propose to include panels with Z300 as in Fig. R4. In Panels G and I one sees that up to the west coast of North America the response to positive WPD is similar (in both the bottom left and bottom right, high pressure is centered over the Alaska/Vancouver area), but that significant differences (panel H) exist further east, with the low pressure south of Greenland and the high pressure over Europe. This supports our conclusion that the strength of the Atlantic jet modulates the downstream travel of the Rossby wave response.

[Figure]

Fig. R4. New version of Figure 7. Bottom row now displays z300 anomalies. Middle column shows only the significant differences (conservative block bootstrapping, as per major comment 4): Modulation of the teleconnection when the west Pacific dipole in week -3 to 0 is in its positive phase. Left column: samples with positive t2m response in week 3 to 6, meaning a positive phase teleconnection (n = 621). Right column: samples resulting in a neutral or negative t2m response, meaning an absence of the teleconnection (n = 645). Middle column: difference between panels in the left and right column. Values found to not exceed the difference as expected by random chance are masked (α = 0.1, more detail in Section 5). Top row: composite anomalies of SST in week -3 to 0. Middle row: composite U300 anomalies in week -3 to 0. Contour-overlay shows the summertime climatological U300 value. Bottom row: composite Z300 anomalies in week 1 to 2. Composites are extracted from the period 1980-2021 (climate normal also defined from 1980-2021).

L281-284: Are there any evidences supporting that the more frequent warming & high pressure over Eastern Europe and Russia are statistically significant related to the increasing WPD negative phase? The evidence provided in this study is indirect, basically because the reasoning consists of a combination of Figure 4 and Figure 5. Figure 4 provides evidence that the link to western Europe has become statistically significant. Figure 5 shows that the surface imprint of the teleconnection pattern is opposite for Western Europe and Eastern Europe and Russia. This is now clearer because we've added the t2m in that region to the composite plots, as per minor comment L208-209. We however consider it out of scope for the current manuscript to provide direct evidence by repeating the analysis of statistical significance behind Figure 4, but with a different, Russian response variable.

**REVIEWER 2**

The study by van Straaten et al investigates the link between sub-seasonal variability in tropical Pacific convection, closely related to ENSO, and summertime circulation over the Euro-Atlantic. The study shows that SST gradients associated with the dipole represent a combination of ENSO variability and West Pacific warming. These gradients in the West Pacific are followed by quasi-stationary waves that linger for multiple weeks. Situations with La Niña-like gradients are followed by high-pressure centers over Eastern Europe and Russia, three to six weeks later. The study also confirms earlier findings that the summertime connectivity between the Pacific and Europe has shifted in recent decades and partly explains the increased occurrence of high sea level pressures and summer temperatures over the European continent. The evidence the authors show is compelling and extends

to sub-seasonal scales the findings reported by other authors who have analyzed teleconnections from the Pacific towards Europe. I think the study is worth publishing after minor amendments to the text and figures.

We thank the reviewer for reviewing the manuscript and making helpful suggestions.

Minor comments:

Line 44: There is a typo in consequence.
Fixed

Lines 69-71 Even when a two weeks lag has been used, I consider it important to show that this is the optimal time lag since it is not so obvious that the signals arrive with the same lag.

This links to the question of whether this is a sub-seasonal or seasonal/concurrent phenomena (major comment 5 of Reviewer #1).

We agree that this is indeed good to show. We propose to do so with plot R5, which we place in an appendix. We see that strong relations are found when looking at concurrent near-seasonal averages (9-12 weeks, panel B). But equally strong or even stronger relations are also found for shorter timescales of 8 to 4 weeks (panel A), in which case the strongest link is not found concurrently, but exists when the variables are related with a separation of zero to one week (between the end of the WPD period and the start of the t2m period). That a certain separation is required is even clearer when we follow the correlation values horizontally along the 2-week aggregation (second row from the bottom). These values peak at lags of 2 and 3 weeks, which corresponds to the common finding in literature that Rossby waves from tropics need two weeks to fully establish in the mid-latitudes (see also Major comment 2 of Reviewer #3). Overall this leads us to conclude that the link we are studying in this manuscript could be found in concurrent seasonal aggregates, but that this signal consists of an underlying lagged sub-seasonal connection. And with the hindsight of this new figure we can conclude that the timescales, as informed by our previous study, and as used in the presented analysis (4-week aggregates, and a 2-week separation) are at least in the vicinity of the optimum.

We discuss this briefly in the section on the correlation map, which is the first result using these timescales. L128:

*"We correlate four-week-average SST anomalies ('SST in week -3 to 0') to lagged European four-week-averaged t2m in week 3 to 6. The gap of two weeks between the two periods corresponds to the time window over which tropical Rossby Waves still affect the mid latitudes (Branstator, 2014) (alternative lags are discussed in Appendix A)"*

This refers to the new appendix where we include Figure R5.

[Figure]

Fig R5 A) Pearson correlation between WPD and detrended Western European t2m at different lags and levels of temporal averaging (averaging applied equally to both WPD and t2m). Lag is defined as the number of intervening weeks (after the end of WPD and before the start of t2m response). The black cross highlights the original lag and level of temporal averaging chosen in this study. The black square highlights the combinations presented in Figs. R7 and R8. Correlation is computed using data from 1980-2021. B) Correlation obtained when WPD and t2m are fully overlapping (i.e. concurrent).

Line 86-87: A clear reason should be stated on why the different times for averaging the different variables are used. The authors provide citations to other works, but still in my view, for clarity's sake, the authors should mention in this paper the reason for using different time lengths for averaging. The reason for the differing timescales was the fact that we used precomputed monthly indices for relative ENSO and PDO. However, we have now recomputed all results using a 4-week ('monthly') aggregation for our manually defined SST indices (WPD and WNP), which matches the timescales of ENSO and PDO, and also that of the average t2m response in week 3 to 6.

Figure 4B. It would be better to show the result in frequency, meaning the occurrences are divided by the total amount of occurrences and not occurrences. Good idea. Though at least for panel C we do prefer to keep counts, as this gives an idea how many samples are present. These numbers will later match the number of samples going into each composite, presented in Figure 5 and 7. The proposed new plot is Fig R6. We propose to also use the fraction of total amount in the original Fig. 8. See the new Fig R1.

[Figure]

Fig R6. New version of Fig 4 with frequencies instead of absolute counts for panels A and B, and with consistent temporal aggregations (4-week wpd, 2-week gap, 4-week t2m).

**REVIEWER 3**

The paper by Van Straaten et al. studies the connectivity between the western Pacific and Western Europe, with a focus on the sub-seasonal predictability of summer European temperatures related to tropical Pacific sea surface temperatures (SSTs) as well as the modulation of the teleconnection by north Atlantic SSTs. The authors base their analysis on the ERA5 reanalysis and other observed sea surface temperature datasets.

I have a few comments that need to be addressed before a possible publication of the paper in Weather and Climate Dynamics. We thank the reviewer for reviewing the manuscript and making helpful suggestions.

**Major Comments:**

1. My first comment is that the study is based on one reanalysis only (ERA5) while other reanalyses do exist and could be used to assess the sensitivity of the results to the choice of

ERA5. Given that they use the 1950-2021 period for ERA5, I am also assuming that the authors have used the preliminary ERA5 back-extension (1950-1978) and not the final ERA5 recent release since 1940. If it is not too cumbersome, I would suggest to update the analysis with the new ERA5 data (at least for the figures using years before 1979).

The study is indeed based on the preliminary back-extension of ERA5. According to the documentation https://confluence.ecmwf.int/display/CKB/The+family+of+ERA5+datasets the preliminary back-extension could suffer from too intense tropical cyclones. As we do not study these, we think that there is no reason for concern, and that differences will be negligible. Also, the redownloading of all data is quite cumbersome on the IT infrastructure that we employ, which is why we prefer not to extend to 1940.
We acknowledge that other reanalyses do exist and that these could be used to check the sensitivity to the data product. For aspects of this study this has already been done, such as for the west-pacific warming mode (Funk and Hoell, 2015), and the tropical central pacific cooling (Seager et al, 2022), meaning we can be confident that the increase of strong 4-week gradients as captured by negative WPD is robust. Nonetheless we acknowledge the limitation and already included a comment about quality of reanalysis data L341:
*"We should however be cautious about concluding that the teleconnection has been absent before 1980, as it can also relate to data quality which improved with the advent of satellite observations (Hersbach et al., 2020)".*

2. In addition to spatial aggregation for the predictand, the authors use a monthly (4 weeks) aggregation for t2m and 3 weeks for SSTs. It would be interesting to see the results with a shorter temporal aggregation (10-15 days for instance, as the authors explicitly mentioned this period in their introduction as the time needed for the waves to establish) and comment on the potential differences. If these long averaging periods are required to obtain significant results (albeit with low amplitude for the correlations, 0.15–0.2), it is interesting information.

We agree that the sensitivity to temporal aggregation is interesting to test. See our response to the suggestion of Reviewer #2 (Line 86-87). In that response we state that we now include Figure R5 in the appendix. This figure displays that a two-week aggregation actually leads to less prominent correlation, but that at this two-week aggregation it is even clearer that one needs to employ a sub-seasonal (2-week) separation between SST and t2m response to find the clearest connection. This aligns well with the period needed to establish mid-latitude Rossby wave responses, as mentioned in the introduction of the manuscript.

3. Statistical significance: no statistical significance of the composites is provided for figures 5 and 7. They need to be performed with an appropriate method (for a cautionary note about t-test, see "The Use of t values in Composite Analyses" by Brown, T.J., and Hall, B.L., Journal of climate, vol. 12. 2941-2944, 1999). It would be nice to have more details in the paper about the method used to derive statistical significance for Figure2 (without having to go to a previous publication). In relation to the false discovery rate procedure, it is not clear to me what alpha (page 6, line 119) is (in Wilks BAMS2016, alpha is the nominal level of the statistical test).

Significance testing has been added to composite plots. As an appropriate and conservative method we chose to do block-bootstrapping. See our response to major comment 4 of Reviewer #1
The meaning of alpha is indeed the nominal level of the statistical test. In this case we applied very small values of alpha because of the dependence introduced by rolling window averaging. This dependence between samples namely leads to very significant results (when using conventional levels like alpha = 0.05). Dependent on the amount of averaging we thus had to vary the strictness of testing. Empirically we found in the previous study (van Straaten

et al. 2022) that this specific alpha (5*10^-12) worked well for filtering 4-week rolling-average data. We now mention this reason in text, and refer to the previous work for more details. L135:

*"Significance of the partial correlation is determined per grid cell by a two-sided test with nominal level α = 5 · 10−12 (small α is needed because of the dependence introduced by rolling window averaging). After this we apply a false discovery rate correction (Benjamini and Hochberg, 1995) (details can be found in van Straaten et al., 2022)"*

4. Figure 2 and the definition of the WPD index: as it is now, there seems to be a lot of subjectivity in the choice of the two boxes (comp_1 and comp_2). For instance, one could have extended the comp_1 westward and southward (including regions with significant correlations). Similarly, the comp_2 box could have been slightly extended eastward and southward. Is it possible to present a rationale for the choice of the boxes? Or at least, one would like to see a sensitivity analysis regarding the definition of the WPD index. It is a bit surprising to notice that a large spatial aggregation is performed for the predictand (t2m in Europe), contrasting with the small spatial extent of the regions used to estimate the WPD index.

Figure 2 gives the impression that the boxes could indeed have been extended to include some more significantly correlated cells. However, whether these extra cells outside of the boxes show up, depends on the exact chosen timescales and lags. In Fig. R7 we illustrate how the correlation patterns can shift. Influential is also whether one performs the stratified cross-validated correlations on the whole reanalysis period (shown) or only on the last few decades (Fig. R8). We made a subjective choice for boxes that seemed to capture those cells that are relatively consistent across aggregation timescales, lags and subsets of the data, i.e. we chose the boxes such that they would capture 'the robust centres of the phenomenon'. In a subjective sense this means that the presented analysis is one of the more robust options among all possible placements of the boxes. Objectively it is a fact that the SSTs in these boxes were the most useful predictor in a previous study (van Straaten et al. 2023, something we already mention in the manuscript). And it is also a fact that in the current diagnostic framework this definition of WPD works, because links to Europe are found that are in agreement with previous studies. Still we want to acknowledge that alternatives could potentially also work. We now include Fig R7 and R8 in an appendix to illustrate how correlation patterns can shift when different choices are made. In this way the reader can form an opinion about alternative definitions of the index. In the text of section 3 where the boxes are introduced, we discuss the influence of aggregation and lag on potential placement, and refer to the appendix. L161:
*"To test the sensitivity of choices regarding the location of the two boxes, we present additional results in Appendix A. These are additional crossvalidation maps similar to Fig. 2B and show that the exact extent and location of the robustly correlated pattern can shift when different combinations of timescales and lags are chosen. The current boxes are positioned such that only the features shared among multiple combinations are captured."*

[Figure]

Figure R7 As Figure 2B, with partial correlation computed on reanalysis data from 1950 till 2021 (5-fold cross validation). Panels illustrate different combinations of lags and time aggregations to assess alternatives for box placement. The results of this study are based on the aggregation and lag highlighted in red.

[Figure]

Figure R8 As Figure R6, but with partial correlation computed on data from 1980 till 2021 (5-fold cross validation). The results of this study are based on the aggregation and lag highlighted in red.

**Minor comments**:

- Page 2, line 34: Many studies suggest that the contribution of ocean SSTs (including therefore La Niña) to the Russian heatwave is very weak (see Dole et al., 2011; Hauser et al. 2016; Wehrli et al. 2019).

  This is indeed an interesting contrast with the cited study of Schneidereit et al. (2012). Perhaps one way to understand the conundrum is that these three are modeling studies, which, as we also highlight in the introduction, can have problems simulating this type of teleconnection. The study of Schneidereit on the other hand uses reanalysis. Also, the small contribution in the case of Wehrli et al. (2019) might be a consequence of the experimental protocol, which derives SST's influence by comparing to a small collection of reference states that are neighboring in time and perhaps not so different. But nonetheless, we acknowledge that the issue is not settled. We cite the mentioned studies, and state that the influence of the La Nina on the russian heatwave is debated. L34:

  "*It is for instance thought that a (developing) La Niña episode supported the prominent blocking that was part of the Russian heatwave of 2010 (Schneidereit et al., 2012). On the other hand, it is also thought that SSTs had little influence on the Russian heatwave (Dole et al., 2011; Hauser et al., 2016; Wehrli et al., 2019).*"

- Page 2, line 51: this is a reference to a paper under review, I am not sure about the WCD policy about that.

  The paper is now accepted.

- Page 4, line 100: using this PDO definition where global mean SSTs are removed might not be the best option This assumes that any anthropogenic influence on PDO behavior is fully removed by subtracting the time evolving changes in global mean SSTs from local SST changes in the PDO region. This is unlikely as it is known that there are different forced warming rates between the PDO region SSTs and the global mean SSTs. This means that an anthropogenic signal is likely to be aliased when using this PDO index definition (see Bonfils and Santer, 2011).

  Thank you for this insight. This definition was employed by the institute from which we obtained the data: https://oceanview.pfeg.noaa.gov/erddap/tabledap/UW-JISAO_PDO.html But we notice now that the updating of this data is also discontinued. So instead we replace it with a PDO index that does not suffer from this issue https://oceanview.pfeg.noaa.gov/erddap/tabledap/cciea_OC_PDO.html and update the data description accordingly. L111:

  "*The pre-computed index is provided by NOAA and is also based on ersstv5.*"
  Replacing the index hardly changes the presented correlation coefficients in Fig. 3, as PDO remains uncorrelated to the response.

- Page 4: there is no reference to figure 1A, perhaps add it line 109.

  Good suggestion. L125

- Page 6, lines 126–129: I think that this statement (that the WPD captures the combined heating contrast …) need a bit more details, or maybe even an additional figure. It is written as if it is an obvious fact, but it is not.

  True. With combined heating contrast we mean a contrast that is not fully zonal and not fully meridional, but a combination of both. In the original manuscript we highlighted only the main mechanism for the Walker-circulation-type zonal contrasts, namely ENSO. We now expand this text in the introduction to also highlight what we mean by the mechanisms resulting in more meridional contrasts. L38:

  "*Besides the zonally oriented dipole implicated in ENSO, namely between the tropical central Pacific and Maritime continent, meridional orientation is important as well. Diagnosis by Ding et al. (2011) of heating patterns that steer mid-latitude flow reveals that anomalous subtropical heating can relate to QSRWs. Such meridional features reflect activity of the Indian Summer Monsoon and the western north Pacific monsoon, as both consist of anomalous convection extending northward, into the Indian subcontinent and the western north Pacific (WNP) respectively.*"
  We also expand our explanation in section 3 describing the WPD index L152:
  "*Furthermore, with component 2 located in the WNP and component 1 at the eastern edge of*

*the central Pacific, their combination appears to capture a heating contrast with both meridional and equatorial orientation (i.e. the emphasized signal consists of more than just zonally opposing anomalies along the equatorial axis)."*

Furthermore, the statistically significant SST pattern captured by WPD is now clearly shown in Figure 5B. This is discussed in the Section 5, and in words again in the discussion. L322: *"The index captures a pattern of opposing SST anomalies in the central-to-west equatorial Pacific and the Western North Pacific region, i.e. a dipole with both equatorial and meridional orientation, which was shown to relate to changing patterns of deep convection and large-scale SST changes across the Pacific."*

- Page 6, figure 3 caption: why do you use 4 weeks instead of 3 for PDO and Niño indices? This is now harmonized, all variables are at the 4 weeks/monthly timescale. See also comment Line 86-87 of Reviewer #2.

- Page 6, lines 131–134: I do not understand the logic here. The WPD index is based on two boxes that are located in the Western Pacific, and one of the boxes is outside the equatorial region. It is not clear to see how this can be related to the equatorial SST gradient across the whole Pacific basin. Please explain.
  True, the index as defined is not purely based on equatorial gradients, but it is a combination of equatorial and meridional gradients. As mentioned in the manuscript both types were found to be important by Ding et al. (2011). See our response to your minor comment, lines 126–129.

- Page 7, lines 155–159: why not using the "detrending" approach outlined for Figure 2 instead of using a 21-yr rolling window?
  The procedure with the rolling 21-yr window is more involved than detrending with linear regression (the approach taken for Figure 2), but we think it is particularly suited for the data, because of the non-linearity in the long-term t2m warming in the last few decades. The reason we detrend with regression in Figure 2 is that we wanted to simultaneously correct for inflation due to autocorrelation. which is possible when both factors (autocorrelation, and long-term trend) are jointly estimated and removed within a single regression framework. That the approach of Figure 2 is different does not lead to inconsistencies, as after all, the grid-based correlation is only used to inform the placement of the boxes, after which WPD is defined on SST anomalies that are only de-seasonalized. See also our reply to major comment 2 of Reviewer #1.

- Page 7, lines 163–165: please specify what are exactly the equatorial (SST?) gradients you are talking about.
  In this case this is the increase in SST gradient as seen in the long-term warming of the west Pacific (WNP) and the absence of warming in the central Pacific (Nino3). This could have been clearer indeed, also because the gradient is not only equatorial. Additionally we now refer to the long term changes in the 'background state' and call it 'climatological'. L199: *"The increased occurrence of negative WPD phases does agree with WNP warming and the strengthening of climatological gradients over the Pacific (Section 3, Fig. 3A)"*
  Because section 3 has been expanded as per your comment L126–129, the reference here is now more informative.

- Figure 7: the contours are very difficult to see, can you enlarge the plots and make thicker contours?
  Contours are now made thicker. See Fig R4.

**References**:

Bonfils C, Santer BD (2011) Investigating the possibility of a human component in various Pacific Decadal Oscillation indices. Clim Dyn 37:1457–1468. https://doi.org/10.1007/s00382-010-0920-1

Dole, R., Hoerling, M., Perlwitz, J., Eischeid, J., Pegion, P., Zhang, T., et al. (2011).Was there a basis for anticipating the 2010 Russian heat wave? Geophysical Research Letters, 38, L06702. https://doi.org/10.1029/2010GL046582

Hauser, M., Orth, R., & Seneviratne, S. I. (2016). Role of soil moisture versus recent climate change for the 2010 heat wave in western Russia. Geophysical Research Letters, 43, 2819–2826. https://doi.org/10.1002/2016GL068036

Wehrli, K., Guillod, B. P., Hauser, M., Leclair, M., & Seneviratne, S. I. (2019). Identifying key driving processes of major recent heat waves. Journal of Geophysical Research: Atmospheres, 124, 11,746–11,765. https://doi.org/10.1029/2019JD030635

**New author-provided references:**

Deser, C., Terray, L., & Phillips, A. S. (2016). Forced and internal components of winter air temperature trends over North America during the past 50 years: Mechanisms and implications. Journal of Climate, 29(6), 2237-2258.

Faranda, D., Messori, G., Jezequel, A., Vrac, M., & Yiou, P. (2023). Atmospheric circulation compounds anthropogenic warming and impacts of climate extremes in Europe. *Proceedings of the National Academy of Sciences*, *120*(13), e2214525120.

Ossó, A., Sutton, R., Shaffrey, L., and Dong, B. (2020) Development, amplification and decay of Atlantic/European summer weather patterns linked to spring North Atlantic sea surface temperatures, Journal of Climate, 33, 5939–5951, https://doi.org/10.1175/JCLI-D-19-0613.1, 2020.

---

## Author Response (AR2)

**Second response to reviewers for wcd-2023-6**
*Strengthening gradients in the tropical west Pacific connect to European summer temperatures on sub-seasonal timescales*

Comments of Reviewer 2:

I really appreciate the authors carefully answered and addressed all my questions and comments. I think the manuscript is almost ready to publish. The manuscript is well-written, with most of the details are clear. I have some minor comments/suggestions though, hopefully they would be helpful to improve the accessibility of the paper:

We thank the reviewer for reviewing the manuscript again and making further helpful suggestions. Line numbers refer to lines in the tracked-changes manuscript.

L112: It might be helpful to give the warming rate.
We have added a rate estimate, based on the IPCC AR6 WG1 atlas (Guitierrez et al. 2021), and an additional reference (Dong et al. 2017). See L111:
*"In fact, western European summer temperatures have been warming faster than the global average, especially since the 1990's, at a rate of 0.4 to 0.8 ℃/decade (Christidis et al., 2015; Dong et al., 2017; Gutiérrez et al., 2021)."*

L115: Just want to clarify: four-week-average"d" SST anomalies correlated with four-week-averaged t2m -> do you use rolling-window averaged SST and t2m, so for each year, there are 92 four-week-averaged data point (since its 92 days in JJA)?
That is correct. We already mentioned the use of rolling averages in the data section. We repeat it here for clarity. See L115:
*"We correlate four-week-averaged SST anomalies ('SST in week -3 to 0') to the lagged European four-week-averaged response ('t2m in week 3 to 6'). Our use of rolling averages leads to 92 samples per JJA season."*

L119-120: consider rephrasing.
We have expanded the text to more clearly explain the construction of the residuals: L120
*"First we let a linear regression predict observed SST and t2m anomalies using time and the value of the previous time step (details can be found in van Straaten et al., 2022). These predictions are then subtracted from the observed anomalies, resulting in residual SST and t2m. With the confounding effects of global warming and auto-correlation removed, any correlation that remains significant is more likely to represent a sub-seasonal relation."*

L122-125: I think it might be helpful to explain/remind the purpose of this analysis (false discovery rate correction, Fig.2b).
Now explained. L126:
*"To mitigate the accumulation of chance-based discoveries when performing multiple significance tests, we applied a false discovery rate correction (Benjamini and Hochberg, 1995) (details can be found in van Straaten et al., 2022)."*

L129 & fig.2b: component2 box, it seems like the component2 box does not exactly match the high correlation region, the box seems to be westward. It seems like the box selection is a bit of arbitrary.
You are right that the box does not seem to fully overlap with the correlated region. To explain this better, and to contextualize the current placement, we introduced a new set of Appendix results in the last round of revisions. These show that the region of high correlation is influenced by choices of lag, timescale and data-set length. In this sense, determining the box-placement on any one of

those maps can be considered somewhat arbitrary. The current placement tries to encompass features shared by multiple of those maps. We explain this, and refer to the appendix, in the last part of this section. See L149:

*"To test the sensitivity of choices regarding the location of the two boxes, we present additional results in Appendix A. These are additional crossvalidation maps, similar to Fig. 2B, and show that the exact extent and location of the robustly correlated pattern can shift when different combinations of timescales and lags are chosen. The current boxes are positioned such that only the features shared among multiple combinations are captured."*

Fig.3b: what does the shaded colors represent? Also, any statistical significant tests for the trends in Fig.3a and correlations in Fig.3b?

The shaded colors represent the magnitude and sign of the correlation values that are also reported as numbers in panels B and C. Because the numbers themselves are reported we do not think that an additional legend is necessary. However, we expand the caption of Figure 3 to explain the meaning of the shading:

*"Shading in panels B and C illustrates the sign of the reported correlation values (red: positive, blue: negative) and their magnitude (dark: strong, light: weak)."*

Regarding potential significance tests: our goal at this stage is not to prove or disprove an effect but just to highlight potential differences and relatedness among the indices of Pacific variability. One example is the warming of the West Pacific (WNP region) and the absence of warming in the central/eastern tropical Pacific (Nino 4), which is also documented in e.g. Seager et al. 2022 and Wills et al 2022, both of which we cite here. We think that significance tests are therefore not needed.

L160-162: I think the description is not accurate. The change in the climatological background state of the Pacific is "La Nina-like", meaning the western tropical Pacific has been warming fast, while the eastern tropical Pacific has been warming slowly even slightly cooling. This change in SST-pattern can lead to strengthening Walker Circulation. It does not mean the change in the basic-state "interacts with La Nina to produce stronger Walker Circulation".
I understand that this description is from Funk and Hoell 2015 Discussion, but in the more recent studies regarding the trend of the tropical Pacific basic-state (e.g., Seager et al. 2019, 2022; Lee et al. 2022 review paper), they did not mention that the La Nina-like warming basic-state "interacts with La Nina events to produce stronger Walker Circulation".

This language was the result of us trying to accommodate the concerns of Reviewer #1 in the first round of revisions. The reviewer raised that the 'La Nina like' changes in the background state are not entirely interchangeable with the western Pacific warming mode diagnosed by Funk and Hoell 2015. We replied that they indeed have a different spatial emphasis, but that the result is highly equivalent, namely a strengthening of the Walker circulation. However, we do agree with the current concern that the 'interact' language is a bit awkward. We adapt our formulation. L164:

*"These opposing trends reflect documented warming in the western Pacific, while the central to eastern tropical Pacific has not warmed (Wills et al., 2022; Seager et al., 2022; Sun et al., 2022; Lee et al., 2022). The result is that the zonal SST gradient over the tropical Pacific has increased, which is generally referred to as a 'La-Niña-like' change in the Pacific 'background state', and has been linked to a stronger Walker circulation (Lee et al., 2022; Seager et al., 2022). Also the west Pacific warming mode, visible as the warming in the WNP region, enhances the climatological background-gradient over the Pacific, and is linked to a strengthening of the Walker circulation, […]"*

Fig.4: It might be helpful to include a legend in the panel A.

Now included, for both panel A and B. See Fig. 4.

Fig.4 & Section 4: It might be helpful to briefly mention why 21-year rolling window is used here.
We now explain in the text of Section 4 why a rolling window of this length was used: sufficiently short that it allows for non-stationarity in the teleconnection (see e.g. Bahaga 2013 for an investigation of different lengths) but certainly longer than the timescale of ENSO. L184:
*"A window length of 21 summers was deemed sufficiently short to adapt to non-stationarity in the roughly 72-year dataset, but sufficiently long not to be affected by inter-annual variability."*

L278-284: I am not familiar with these studies and the coupling between ocean and jet stream in the North Atlantic, but did these work focus on the same timescales (subseasonal)?
The studies describe a coupling that develops from late-winter and spring into summer. In that sense it is a seasonal phenomenon, because consequences in July and August can be linked to patterns in March and April. In text we already mention this seasonal aspect, which interacts with changes on a shorter timescale such as the month-to-month migration of the jet-stream, and of course also with the sub-seasonal teleconnection from the Pacific (this is why we place the discussion of this phenomenon in the 'modulation' section). To make the seasonal character a bit clearer we amend the text as follows. L288:
*"The tripole pattern spans multiple seasons and can occur already in late winter and early spring. From that moment onward it is known to precede the summertime pattern with a strong low pressure anomaly positioned south of Greenland and west of the British Isles (Fig. 5G)"*
and L298:
*"We therefore deduce that the seasonal interplay of SST tripole and Atlantic jet could modulate the sub-seasonal teleconnection by longitudinally guiding QSRWs from the west Pacific towards Europe."*

L301: stronger "equatorial" and meridional SST gradients: zonal?
That is correct. Changed to 'zonal'. L312

L322: "To a smaller extent also Western European t2m has been increasing" -> consider rephrasing. "smaller extent" -> I am not sure what this means, spatial extent? "increasing" -> warming?
Indeed confusing, this concerns the degree of warming and not spatial extent. We replaced the text L336:
*"The warming of Western European average summer t2m has been less severe, though still highly significant (Christidis et al., 2015; Gutiérrez et al., 2021)."*

L324 & 327: (Fig.8A) -> Figs.4A and 8A. (Fig.8C) -> Figs.4A and 8C
Fig.8: Please consider either to include title in each panel, or change the y-axis. It is not easy to read the figure.
We have simplified the Y-axes for Figure 4 (panels a and b) by including titles and a legend (as per your comment on Figure 4 above.
We also re-arranged the panels in Figure 8 such that y-axes are shared and need only one label. We have added clarifying titles to the columns in Figure 8.

L322-L333: I think these two paragraphs are convoluting. It may be helpful to polish or edit a bit. My understanding is that the authors want to state the discrepancy between "more frequent negative t2m on subseasonal timescales" and "the warming trend of seasonal t2m over Western Europe": if the more frequent negative t2m on subseasonal timescales is due to change in circulations, then there must be other factors (other than the WPD-Europe teleconnections) that cause the warming trend of seasonal t2m.
Your latter comment is indeed what we try to convey. We have now reordered the sentences in these two paragraphs to a more logical structure, and we also adopt your phrasing. L327-345:

*"The emergence of a significant teleconnection in recent decades provides a lens through which recent summer circulation changes over Europe can be interpreted. This is relevant because warming trends are the consequence of multiple factors such as direct forcing by greenhouse gases and aerosols (Dong et al., 2017), but also circulation changes (Deser et al., 2016; Faranda et al., 2023). The changes diagnosed in this study are an increased occurrence of negative WPD and a reduced occurrence of positive WPD, each with a respective QSRW response. The increased frequency of the negative WPD phase (Fig. 4A), would, according to its corresponding QSRW, induce a warming in eastern Europe and Russia (Fig. 5M). Indeed, high pressure has become more prevalent in this region (Lee et al., 2017; Kim and Lee, 2022; Teng et al., 2022), associated with a very strong increase in heat waves (Rousi et al., 2022), with average summer t2m increasing more than in our Western European target region (Teng et al., 2022).*

*The warming of Western European average summer t2m has been less severe, though still highly significant (Christidis et al., 2015; Gutiérrez et al., 2021). A precise quantification of the contribution from changes in WPD to this trend is beyond the scope of this study (and is also hindered by the methodological necessity to isolate the sub-seasonal teleconnection from the trend by using 21-year rolling window distributions of t2m, see section 4). But relative to the trend, two things can be said. First, during the time that negative WPD has roughly doubled in frequency (Fig. 8A), it remained consistently related to the cold Western European t2m tercile (Fig. 8B). This means that Western European warming is likely caused by other factors.*

*Second, one of these factors could be positive WPD. […]."*

L325-326 "This means that if the teleconnection is influencing Western Europe through circulation changes, then its effect among all other factors, would be a dampening of the warming." This sentence confuses me.
We have adapted this paragraph. See our reply above.

Author provided references:

Bahaga, T. K., Fink, A. H., & Knippertz, P. (2019). Revisiting interannual to decadal teleconnections influencing seasonal rainfall in the Greater Horn of Africa during the 20th century. *International Journal of Climatology*, *39*(5), 2765-2785.

Dong, B., Sutton, R. T., & Shaffrey, L. (2017). Understanding the rapid summer warming and changes in temperature extremes since the mid-1990s over Western Europe. *Climate Dynamics*, *48*, 1537-1554.

Gutiérrez, J.M., R.G. Jones, G.T. Narisma, L.M. Alves, M. Amjad, I.V. Gorodetskaya, M. Grose, N.A.B. Klutse, S. Krakovska, J. Li, D. Martínez-Castro, L.O. Mearns, S.H. Mernild, T. Ngo-Duc, B. van den Hurk, and J.-H. Yoon, 2021: Atlas. In *Climate Change 2021: The Physical Science Basis. Contribution of Working Group I to the Sixth Assessment Report of the Intergovernmental Panel on Climate Change* [Masson-Delmotte, V., P. Zhai, A. Pirani, S.L. Connors, C. Péan, S. Berger, N. Caud, Y. Chen, L. Goldfarb, M.I. Gomis, M. Huang, K. Leitzell, E. Lonnoy, J.B.R. Matthews, T.K. Maycock, T. Waterfield, O. Yelekçi, R. Yu, and B. Zhou (eds.)]. Cambridge University Press, Cambridge, United Kingdom and New York, NY, USA, pp. 1927–2058